# Low chorionic villous succinate accumulation associates with recurrent spontaneous abortion risk

Xiao-Hui Wang [1,2,6], Sha Xu [2,3,6], Xiang-Yu Zhou[1,6], Rui Zhao[2], Yan Lin[2,3], Jing Cao[4], Wei-Dong Zang[4], Hui Tao[5], Wei Xu [2,3], Ming-Qing Li[2], Shi-Min Zhao [2,3], Li-Ping Jin[1✉] & Jian-Yuan Zhao [1,2,3✉]

Dysregulated extravillous trophoblast invasion and proliferation are known to increase the risk of recurrent spontaneous abortion (RSA); however, the underlying mechanism remains unclear. Herein, in our retrospective observational case-control study we show that villous samples from RSA patients, compared to healthy controls, display reduced succinate dehydrogenase complex iron sulfur subunit (SDHB) DNA methylation, elevated SDHB expression, and reduced succinate levels, indicating that low succinate levels correlate with RSA. Moreover, we find high succinate levels in early pregnant women are correlated with successful embryo implantation. SDHB promoter methylation recruited MBD1 and excluded c-Fos, inactivating SDHB expression and causing intracellular succinate accumulation which mimicked hypoxia in extravillous trophoblasts cell lines JEG3 and HTR8 via the PHD2-VHL-HIF-1α pathway; however, low succinate levels reversed this effect and increased the risk of abortion in mouse model. This study reveals that abnormal metabolite levels inhibit extravillous trophoblast function and highlights an approach for RSA intervention.

[1] Clinical and Translational Research Center, Shanghai First Maternity and Infant Hospital, Tongji University School of Medicine, Shanghai, China. [2] Institute of Metabolism and Integrative Biology, State Key Lab of Genetic Engineering, School of Life Sciences, Obstetrics & Gynecology Hospital of Fudan University, Key Laboratory of Reproduction Regulation of NPFPC, and Zhongshan Hospital of Fudan University, Fudan University, Shanghai, China. [3] Collaborative Innovation Center for Genetics and Development, Institutes of Biomedical Sciences, Fudan University, Shanghai, China. [4] Department of Anatomy and Neuroscience Research Institute, School of Basic Medical Sciences, Zhengzhou University, Zhengzhou, China. [5] Second Hospital of Anhui Medical University, Anhui Medical University, Hefei, China. [6] These authors contributed equally: Xiao-Hui Wang, Sha Xu, Xiang-Yu Zhou. ✉email: jinlp01@163.com; zhaojy@fudan.edu.cn

R ecurrent spontaneous abortion (RSA) is a highly heterogeneous condition defined as three or more consecutive spontaneous abortions before 20 weeks of gestation and affects 1–2% of fertile women worldwide.[1] Although etiological studies have reported various factors that affect RSA, including chromosomal and anatomical uterine abnormalities, endometrial infections, endocrine abnormalities, antiphospholipid syndrome, inherited thrombophilias, alloimmune causes, and genetic factors,[2–4] its underlying mechanism remains unclear and a large proportion of RSA cases remain unexplained.[1,5,6]

Embryonic development requires placenta formation following blastocyst implantation in the maternal organism. The human placenta is characterized by the extensive invasion of trophoblasts into the maternal uterus at the maternal–fetal interface between the uterine mucosa and the extraembryonic tissues of the developing conceptus, allowing direct contact between trophoblasts and maternal blood.[7,8] Abnormal placentation and incomplete formation of the maternal–fetal interface in the first trimester are believed to cause RSA.[9–12] After the initial nidation phase, human trophoblasts differentiate along either the villous or extravillous trophoblast pathways.[6,9] Villous trophoblasts cover chorionic villi and are involved in gas and nutrient exchange between the mother and fetus, whereas extravillous trophoblasts invade deep into the uterine wall (as far as the myometrium) and are directly implicated in the anchoring of chorionic villi. During the first trimester, extravillous trophoblasts display high proliferation and invasion like malignant tumor cells; however, trophoblasts rarely become malignant or immortalized even if they disperse through the host vascular system. Proper trophoblast invasion and proliferation are crucial to embryonic implantation and placental development, which promote the establishment of an appropriate maternal–fetal relationship.[2,13,14] The invasive/proliferative properties of human extravillous trophoblasts are known to increase during the first trimester and decrease following embryo implantation under physiological conditions yet decrease abnormally under pathological conditions; however, despite their importance, the underlying molecular mechanisms have been poorly characterized.

Early placental development occurs in a predominantly low-oxygen environment and is, at least partially, under the control of hypoxia signaling pathways.[15] Trophoblast invasion and proliferation are critically regulated by oxygen, while low-oxygen levels direct the differentiation of trophoblast stem cells toward a phenotype associated with the junctional zone[16–18] via a developmental process dependent upon the hypoxia-inducible factor (HIF) signaling pathways. During the first trimester of pregnancy, the human placenta develops in a hypoxic environment caused by the occlusion of uterine spiral arterioles by extravillous trophoblasts. Before week 12 of pregnancy, relative low-oxygen tension is crucial for a successful pregnancy and can promote EVT invasion and differentiation.[19,20] Once the vascular remodeling is accomplished, oxygen becomes vital for normal development. Hypoxia at this stage, which can most often happen with preeclampsia after week 20, does affect placenta function.[21] Although it is well-established that hypoxia plays a role in regulating extravillous trophoblasts function, the underlying mechanism remains unclear. Intracellular oxygen levels are sensed by the prolyl hydroxylase (PHD) family, which regulates downstream hypoxia effectors, such as HIFs. Given that PHD family oxygen sensing requires the co-substrate α-ketoglutarate (α-KG), an intermediate in the citrate cycle, dysregulated glucose and amino acid metabolism may contribute to the onset of RSA.

Extravillous trophoblasts display high proliferation and invasion during the first trimester; therefore, we hypothesized that metabolites and metabolic enzymes may play important roles in modulating trophoblast characteristics. We compared the expression and concentration of glycolytic and citrate cycle metabolites and enzymes in villi and decidua from individuals with or without RSA during different stages of pregnancy. We found that succinate can maintain hypoxia and extravillous trophoblast function, with dysregulated succinate metabolism contributing toward RSA onset.

## Results

**Increased SDHB activity decreases villous succinate levels in patients with RSA.** To investigate whether metabolite profiles were associated with the risk of RSA, we collected 30 pairs of villous and decidua tissue samples from the placentas of patients with RSA and those with clinically normal pregnancies which had been terminated for nonmedical reasons. Metabolic profiling was performed on the samples using nuclear magnetic resonance (NMR), successfully quantifying 31 different types of metabolites in >95% of the samples, involving glucose, amino acids, nucleotides, and fatty acids (Supplementary Fig. 1a). Among these metabolites, we found that succinate levels were 27.1% lower in the villi of patients with RSA than in the control group, while fumarate levels were 19.5% higher and lactate levels were 14.4% lower (Table 1). No significant differences in metabolite levels were observed between the decidua of individuals with RSA and the controls (Supplementary Table 1), suggesting that metabolic alterations in villi, but not decidua, are correlated with the occurrence of RSA.

Since glycolysis and citrate cycle enzymes determine the balance of metabolites, we also compared the expression levels of 20 glycolysis and citrate cycle proteins in villous tissue samples from six patients with RSA and six controls. Succinate dehydrogenase complex iron sulfur subunit B (SDHB) protein levels were significantly elevated in the samples from patients with RSA (Fig. 1a); therefore, we investigated the mRNA expression levels of the genes encoding the same 20 glycolytic and citrate cycle proteins in both RSA and control villous tissue samples using qRT-PCR. SDHB mRNA levels were significantly higher in the samples from patients with RSA, consistent with its protein levels (Fig. 1b). SDHB is an important succinate dehydrogenase subunit that is involved in complex II of the mitochondrial electron transport chain and is responsible for transferring electrons from succinate to ubiquinone, generating fumarate. To test its effect on SDH and intracellular succinate levels, we overexpressed SDHB in human placenta-derived JEG3 (ATCC HTB-36$^{TM}$) and HTR8 (ATCC CRL-3271$^{TM}$) cells. SDHB overexpression increased SDH activity (Fig. 1c) and decreased intracellular succinate levels (Fig. 1d); whereas its knockdown inactivated SDH (Fig. 1e) and caused succinate accumulation (Fig. 1f). Moreover, we confirmed that the relative SDH activity increased remarkably in RSA villous tissues, but not in decidua tissues, compared to those in the healthy controls (Fig. 1g). Together with the finding that either SDHB overexpression or SDHB knockdown did not change the mitochondrial mass in JEG3 and HTR8 cells (Supplementary Fig. 1b), these results indicate that enhanced SDHB expression decreases succinate levels which correlate with RSA onset through increasing intracellular SDH activity.

**Villous SDHB and succinate levels change dynamically during pregnancy under physiological conditions.** To determine the role of succinate in RSA onset, we examined SDHB and succinate levels during normal pregnancy under physiological conditions. During the first trimester, the successful adhesion and invasion of trophoblasts at the maternal–fetal interface are crucial for placenta formation. Since this process requires complex interactions between villi and decidua, we screened the levels of glycolysis and

**Table 1 Metabolite concentrations in villi from individuals with RSA or normal controls.**

| Metabolites (μmol/g) | Normal (n = 30) | RSA (n = 30) | P value | Adjusted P value |
|---|---|---|---|---|
| Glucose | 3.166 ± 0.893 | 3.257 ± 0.767 | 0.676 | 0.947 |
| Pyruvate | 0.306 ± 0.085 | 0.279 ± 0.09 | 0.251 | 0.828 |
| Lactate | 3.367 ± 1.221 | 2.881 ± 0.921 | 0.087 | 0.828 |
| Citrate | 0.303 ± 0.097 | 0.277 ± 0.085 | 0.282 | 0.828 |
| Succinate | 0.589 ± 0.121 | 0.429 ± 0.109 | $1.48 \times 10^{-6}$ | $5.18 \times 10^{-5}$ |
| Fumarate | 0.146 ± 0.05 | 0.174 ± 0.053 | 0.037 | 0.653 |
| Phenylalanine | 0.226 ± 0.048 | 0.212 ± 0.049 | 0.265 | 0.828 |
| Tyrosine | 0.209 ± 0.044 | 0.208 ± 0.061 | 0.910 | 0.992 |
| Aspartate | 1.598 ± 0.419 | 1.502 ± 0.484 | 0.413 | 0.828 |
| Glutamate | 2.088 ± 0.656 | 1.874 ± 0.684 | 0.221 | 0.828 |
| Glutamine | 2.821 ± 0.722 | 2.659 ± 0.731 | 0.391 | 0.828 |
| Glycine | 1.428 ± 0.589 | 1.367 ± 0.478 | 0.664 | 0.947 |
| Valine | 0.509 ± 0.121 | 0.509 ± 0.145 | 0.992 | 0.992 |
| Isoleucine | 0.273 ± 0.071 | 0.295 ± 0.107 | 0.370 | 0.828 |
| Leucine | 0.507 ± 0.113 | 0.516 ± 0.142 | 0.792 | 0.990 |
| Histidine | 0.182 ± 0.108 | 0.168 ± 0.098 | 0.594 | 0.945 |
| Alanine | 1.291 ± 0.298 | 1.223 ± 0.324 | 0.402 | 0.828 |
| Acetate | 0.294 ± 0.053 | 0.307 ± 0.072 | 0.426 | 0.828 |
| Formate | 0.362 ± 0.145 | 0.415 ± 0.214 | 0.264 | 0.828 |
| Inosine | 0.332 ± 0.192 | 0.315 ± 0.173 | 0.722 | 0.973 |
| Creatine | 1.062 ± 0.59 | 1.023 ± 0.503 | 0.784 | 0.990 |
| Hypoxanthine | 0.683 ± 0.233 | 0.63 ± 0.236 | 0.392 | 0.828 |
| Carnosine | 0.019 ± 0.012 | 0.019 ± 0.011 | 0.983 | 0.992 |
| Uridine | 0.338 ± 0.16 | 0.342 ± 0.17 | 0.932 | 0.992 |
| Taurine | 4.351 ± 2.288 | 3.778 ± 1.605 | 0.266 | 0.828 |
| 3-hydroxybutyrate | 0.366 ± 0.304 | 0.371 ± 0.317 | 0.946 | 0.992 |
| Nicotinamide | 0.113 ± 0.071 | 0.11 ± 0.061 | 0.857 | 0.992 |
| NAD | 0.024 ± 0.011 | 0.021 ± 0.012 | 0.317 | 0.828 |
| AMP | 0.009 ± 0.008 | 0.009 ± 0.008 | 0.935 | 0.992 |
| ADP | 0.017 ± 0.011 | 0.016 ± 0.01 | 0.594 | 0.945 |
| Uracil | 0.25 ± 0.056 | 0.235 ± 0.068 | 0.355 | 0.828 |

*NAD* nicotinamide adenine dinucleotide, *AMP* adenosine monophosphate, *ADP* adenosine diphosphate.
Data presented are given in μmol/g (mean ± SD). *P* values were derived from unpaired two-sample *t* test (two groups have the same SD) or unpaired two-sample *t* test with Welch's correction (two groups do not have the equal SD) and then corrected by using the Benjamini & Hochberg method for multiple sampling (adjusted *P* value). Source data are provided as a Source Data file.

citrate cycle enzymes and metabolites in both villous and decidua tissue samples from individuals with normal pregnancies that had been terminated for nonmedical reasons. The majority of glycolytic and citrate cycle enzymes displayed higher expression levels in villi than in decidua, except for SDHB and PGK1 (Fig. 2a), while most glycolytic and citrate cycle enzyme-encoding genes were transcriptionally activated at the RNA level, except for SDHB and PGK1 which were transcriptionally silenced (Fig. 2b). These results indicate that under physiological conditions, villi display higher glucose metabolism than maternal tissue; however, a decrease in SDHB was observed, which may cause further succinate accumulation. Metabolite analysis revealed that most metabolites were present at significantly higher levels in villi than in decidua, with succinate levels increasing the most significantly (Table 2).

Next, we explored SDHB expression during different stages of pregnancy. Interestingly, SDHB protein (Fig. 2c) and mRNA (Fig. 2d) expression were both lower in villi than in decidua during the early stage of embryo implantation (samples collected from days 43 to 70); however, SDHB expression increased significantly to almost the same level as in decidua after the second trimester (samples collected from days 93 to 141) (Fig. 2c, d). The expression levels of other glycolytic and citrate cycle enzymes did not change from the first to second trimesters (Supplementary Fig. 1c). We also examined metabolite levels, confirming that villous succinate levels were high during the early stages of pregnancy and then decreased to levels comparable to those in decidua after the second trimester (Fig. 2e). Taken together, these results indicate that low SDHB expression and high-succinate levels are important for maintaining embryo implantation in the first trimester; therefore, high SDHB expression and low-succinate levels during early pregnancy may increase the risk of RSA.

**Villous SDHB transcription is dynamically regulated by DNA methylation.** To investigate why villous SDHB expression increases pathologically in RSA but exhibits dynamically low levels during the first trimester followed by high levels under physiological conditions, we used the online software "jaspar" (http://jaspar2016.genereg.net/cgi-bin/jaspar_db.pl) to analyze potential *SDHB* transcription factors. CREB1, c-Fos, NR4A2, and RHOXF1 had potential binding sites in the *SDHB* gene promoter region (Supplementary Fig. 2a); however, no changes were observed in their levels in the villi of normal healthy controls and patients with RSA (Supplementary Fig. 2b). Besides, we sequenced the promoter region of *SDHB* gene in villi from 10 normal pregnant women and 20 RSA patients and found no mutations correlated with RSA according to existing literature reports (Supplementary Table 2).

Since epigenetic alterations can also regulate gene expression, we detected CpG site methylation levels in the *SDHB* promoter region of villi from both individuals with RSA and controls (Fig. 3a and Supplementary Fig. 2c). DNA methylation levels were significantly lower at the −1185, −1172, −1165, −1135, and −1077 CpG sites in the *SDHB* promoter of villous samples from patients with RSA (Fig. 3a), consistent with the increased SDHB expression observed in these samples (see Fig. 1b, c). We also found that DNA methylation at the −1172, −1165, and −1135

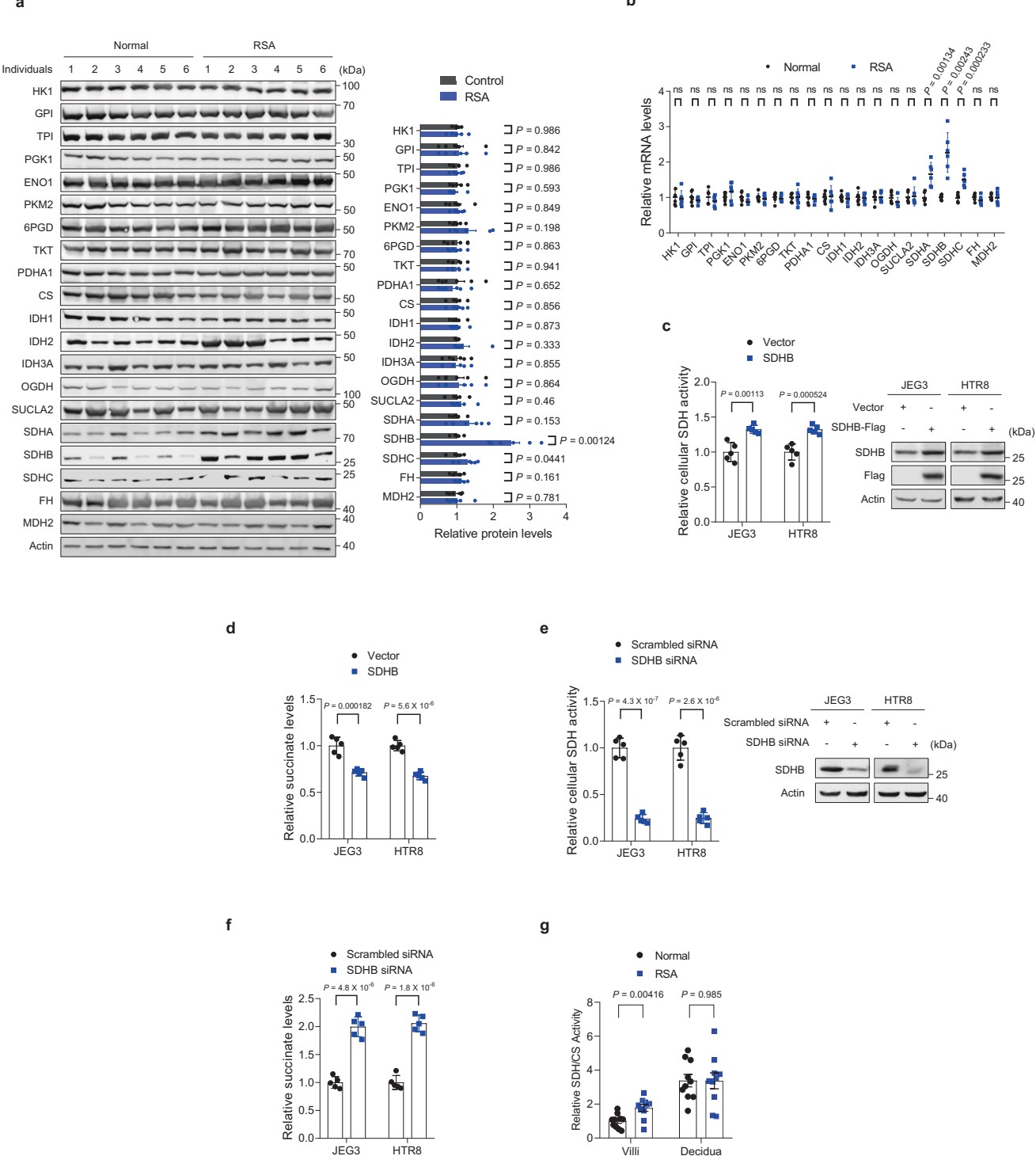

**Fig. 1 Low villous SDHB expression reduces succinate accumulation and correlates with RSA. a** Western blot analysis of glycolysis and citrate cycle enzymes in villi from individuals with RSA or normal controls ($n = 6$ persons/group). Right panel: quantitative western blot results of six samples per group. **b** mRNA expression of glycolytic and citrate cycle genes in villi from individuals with RSA and normal controls ($n = 6$ persons/group). Exact $P$ values in **b** are listed below: HK1 0.800, GPI 0.326, TPI 0.369, PGK1 0.202, ENO1 0.145, PKM2 0.685, 6PGD 0.715, TKT 0.914, PDHA1 0.691, CS 0.841, IDH1 0.546, IDH2 0.801, IDH3A 0.969, OGDH 0.163, SUCLA2 0.816, SDHA 0.00134, SDHB 0.00243, SDHC 0.000233, FH 0.222, and MDH2 0.938. **c** Relative SDH activities in SDHB-overexpressing JEG3 and HTR8 cells and control cells ($n = 5$ independent experiments). The SDHB overexpression efficiency is shown in the right panel. **d** Succinate levels in SDHB-overexpressing JEG3 and HTR8 cells and control cells ($n = 5$ independent experiments). **e** Relative SDH activities in SDHB-knockdown JEG3 and HTR8 cells and control cells ($n = 5$ independent experiments). The SDHB-knockdown efficiency is shown in the right panel. **f** Succinate levels in SDHB-knockdown JEG3 and HTR8 cells and control cells ($n = 5$ independent experiments). **g** The relative SDH activities in villi and decidua tissues from both RSA patients and normal controls ($n = 10$ persons/group). **a–g** Data represent the mean ± standard error; ns not significant, *$P < 0.05$, **$P < 0.01$, ***$P < 0.001$, ns not significant. Two-tailed unpaired Student's $t$ tests in **a–g**. Source data are provided as a Source Data file.

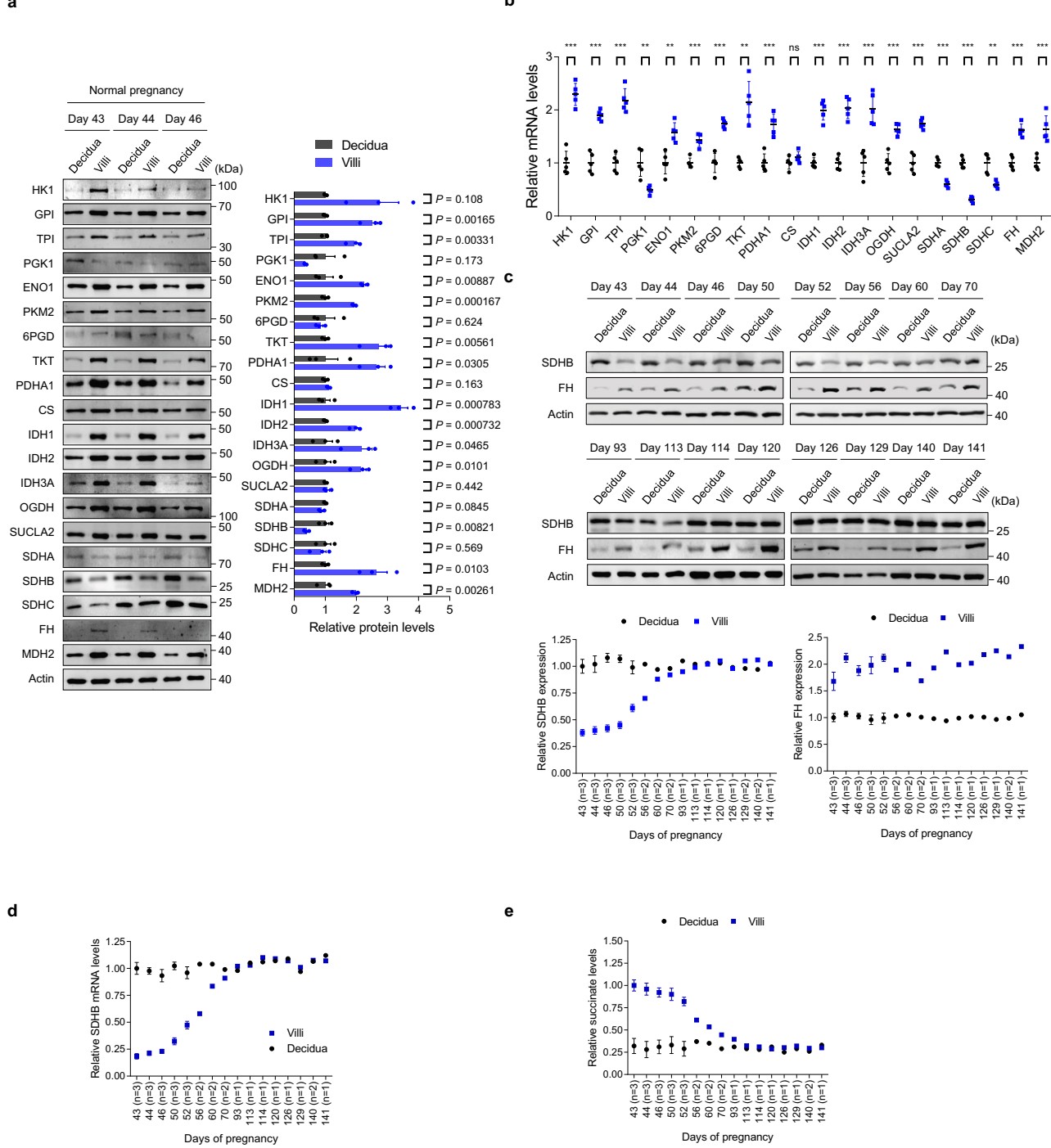

**Fig. 2 Dynamic changes in SDHB regulate villous succinate levels and pregnancy progression. a** Western blot analysis of glycolysis and citrate cycle enzymes in villi ($n = 3$ biologically independent samples) and decidua ($n = 3$ biologically independent samples) from normal controls. The timing of pregnancy termination (day) is shown for each sample. Right panel: quantitative western blot results of three samples per group. **b** mRNA levels of glycolytic and citrate cycle genes ($n = 5$ persons). Exact $P$ values in **b** are listed below: HK1 $1.3 \times 10^{-5}$, GPI $1.4 \times 10^{-5}$, TPI $1.3 \times 10^{-5}$, PGK1 0.00247, ENO1 0.00142, PKM2 0.000206, 6PGD $3.4 \times 10^{-5}$, TKT 0.00225, PDHA1 0.000238, CS 0.164, IDH1 $5.9 \times 10^{-6}$, IDH2 $8.5 \times 10^{-6}$, IDH3A 0.000383, OGDH 0.000115, SUCLA2 $2.8 \times 10^{-5}$, SDHA $1.8 \times 10^{-5}$, SDHB $6.4 \times 10^{-7}$, SDHC 0.00162, FH $4.6 \times 10^{-5}$, and MDH2 0.000902. **c** SDHB and FH expression during pregnancy progression. The timing of pregnancy termination (Day) is shown for each sample. Lower panel: quantitative western blot results. **d** SDHB mRNA levels in villi and decidua during pregnancy progression. **e** Succinate levels in villi and decidua during pregnancy progression. The number of samples at each pregnancy termination ($n$) is shown on the horizontal coordinate (**c–e**). **a**, **b** Data represent the mean ± standard error, *$P < 0.05$, **$P < 0.01$, ***$P < 0.001$, ns not significant. Two-tailed unpaired Student's $t$ tests in **a**, **b**. Source data are provided as a Source Data file.

**Table 2 Metabolite concentrations in villi and decidua from normal pregnancies.**

| Metabolites (μmol/g) | Decidua ($n = 29$) | Villi ($n = 30$) | *P* value | Adjusted *P* value |
|---|---|---|---|---|
| Glucose | 3.286 ± 0.965 | 3.166 ± 0.893 | 0.624 | 0.809 |
| Pyruvate | 0.279 ± 0.093 | 0.306 ± 0.085 | 0.259 | 0.567 |
| Lactate | 3.004 ± 1.158 | 3.367 ± 1.221 | 0.247 | 0.567 |
| Citrate | 0.263 ± 0.076 | 0.303 ± 0.097 | 0.087 | 0.329 |
| Succinate | 0.465 ± 0.111 | 0.589 ± 0.121 | $1.37 \times 10^{-4}$ | 0.005 |
| Fumarate | 0.159 ± 0.06 | 0.146 ± 0.05 | 0.377 | 0.66 |
| Phenylalanine | 0.198 ± 0.053 | 0.226 ± 0.048 | 0.038 | 0.171 |
| Tyrosine | 0.189 ± 0.055 | 0.209 ± 0.044 | 0.123 | 0.39 |
| Aspartate | 1.404 ± 0.454 | 1.598 ± 0.419 | 0.094 | 0.329 |
| Glutamate | 1.908 ± 0.68 | 2.088 ± 0.656 | 0.304 | 0.626 |
| Glutamine | 2.654 ± 0.833 | 2.821 ± 0.722 | 0.415 | 0.691 |
| Glycine | 1.488 ± 0.564 | 1.428 ± 0.589 | 0.69 | 0.833 |
| Valine | 0.447 ± 0.099 | 0.509 ± 0.121 | 0.036 | 0.171 |
| Isoleucine | 0.251 ± 0.069 | 0.273 ± 0.071 | 0.233 | 0.567 |
| Leucine | 0.464 ± 0.13 | 0.507 ± 0.113 | 0.182 | 0.49 |
| Histidine | 0.136 ± 0.049 | 0.182 ± 0.108 | 0.039 | 0.171 |
| Alanine | 1.128 ± 0.268 | 1.291 ± 0.298 | 0.031 | 0.171 |
| Acetate | 0.281 ± 0.059 | 0.294 ± 0.053 | 0.375 | 0.66 |
| Formate | 0.419 ± 0.17 | 0.362 ± 0.145 | 0.177 | 0.49 |
| Inosine | 0.318 ± 0.151 | 0.332 ± 0.192 | 0.76 | 0.847 |
| Creatine | 1.019 ± 0.494 | 1.062 ± 0.59 | 0.762 | 0.847 |
| Hypoxanthine | 0.672 ± 0.283 | 0.683 ± 0.233 | 0.873 | 0.926 |
| Carnosine | 0.019 ± 0.012 | 0.019 ± 0.012 | 0.959 | 0.97 |
| Uridine | 0.37 ± 0.19 | 0.338 ± 0.16 | 0.489 | 0.713 |
| Taurine | 4.089 ± 1.612 | 4.351 ± 2.288 | 0.614 | 0.809 |
| 3-hydroxybutytrate | 0.308 ± 0.111 | 0.366 ± 0.304 | 0.338 | 0.657 |
| Nicotinamide | 0.126 ± 0.066 | 0.113 ± 0.071 | 0.475 | 0.713 |
| NAD | 0.016 ± 0.007 | 0.024 ± 0.011 | 0.001 | 0.015 |
| AMP | 0.009 ± 0.005 | 0.009 ± 0.008 | 0.774 | 0.847 |
| ADP | 0.016 ± 0.01 | 0.017 ± 0.011 | 0.657 | 0.822 |
| Uracil | 0.211 ± 0.066 | 0.25 ± 0.056 | 0.017 | 0.146 |

*NAD* nicotinamide adenine dinucleotide, *AMP* adenosine monophosphate, *ADP* adenosine diphosphate.
Data presented are given in μmol/g (mean ± SD). *P* values were derived from unpaired two-sample *t* test (two groups have the same SD) or unpaired two-sample *t* test with Welch's correction (two groups do not have the equal SD) and then corrected by using the Benjamini & Hochberg method for multiple sampling (adjusted *P* value). Source data are provided as a Source Data file.

CpG sites in the *SDHB* promoter decreased from the first to the third trimester in healthy control individuals, including seven whose pregnancies were terminated in the first trimester, three whose pregnancies were terminated in the second trimester, and three obtained from individuals who had undergone normal vaginal delivery (Fig. 3b and Supplementary Fig. 2d). In addition, blocking DNA methylation with either decitabine or azacitidine decreased DNA methylation at the −1172, −1165, and −1135 CpG sites in the *SDHB* promoter (Fig. 3c, d), and increased SDHB mRNA (Fig. 3e, f) and protein levels (Fig. 3g) in both JEG3 and HTR8 cells, suggesting that DNA methylation at the −1172, −1165, and −1135 CpG sites of the *SDHB* promoter contributes toward dynamic SDHB expression in villi during embryo implantation and that decreased *SDHB* promoter methylation may cause RSA.

**DNA methylation of the SDHB promoter recruits MBD1 and excludes c-Fos binding.** We observed that −1172, −1165, and −1135 CpG sites of *SDHB* promoter maintained high methylation levels in the villi of healthy pregnancies during the first trimester, with average CpG methylation ratios of 0.76, 0.8, and 0.71, respectively (Fig. 3a). Therefore, we hypothesized that methyl-CpG-binding proteins might bind to these sites and regulate *SDHB* transcription. To examine this hypothesis, we performed RNA sequencing on human villi from normal pregnancies, finding that the majority of methyl-CpG-binding proteins were expressed in villi (Supplementary Fig. 3a). However, among seven methyl-CpG-binding proteins, only the knockdown of the methyl-CpG-binding protein MBD1 increased

SDHB mRNA and protein expression, suggesting that MBD1 may be involved in DNA methylation-mediated transcriptional regulation (Fig. 3h, i and Supplementary Fig. 3b, c). Using in vitro electrophoretic mobility shift assays (EMSA), we confirmed that MBD1 bound to the *SDHB* promoter region around the methylated −1165, −1172, and −1135 CpG sites (Fig. 3j).

Since c-Fos had a predicted binding site in the −1152/−1145 region of the *SDHB* promoter (Supplementary Fig. 2a), which is between the −1172/−1165/−1135 CpG sites, we hypothesized that c-Fos might also play a role in DNA methylation-related *SDHB* regulation. Indeed, we confirmed that c-Fos bound to the −1165/−1135 region using in vitro EMSA (Fig. 3k) and found that both MBD1 and c-Fos bound to the same hypermethylation region (region 6) in JEG3 cells based on chromatin immunoprecipitation analyses (Fig. 3l). The binding ability of c-Fos to *SDHB* promoter increased in the MBD1-knockdown JEG3 and HTR8 cells, compared to respective control cells, suggesting that c-Fos competes with MBD1 for the *SDHB* promoter (Fig. 3m). Importantly, region 6 containing the −1172/−1165/−1135 CpG sites was occupied by both MBD1 and c-Fos in villi from normal pregnant women; however, MBD1 occupation decreased and c-Fos occupation increased when −1172/−1165/−1135 CpG site methylation decreased in the villi of patients with RSA (Fig. 3n). These in vivo findings indicate that under physiological conditions, CpG site hypermethylation in the *SDHB* promoter recruits MBD1 and excludes c-Fos binding, thus avoiding the transcriptional activation of *SDHB*. Conversely, under pathological conditions, decreased *SDHB* promoter methylation excludes MBD1 and recruits c-Fos binding to activate *SDHB* expression and thus induce RSA.

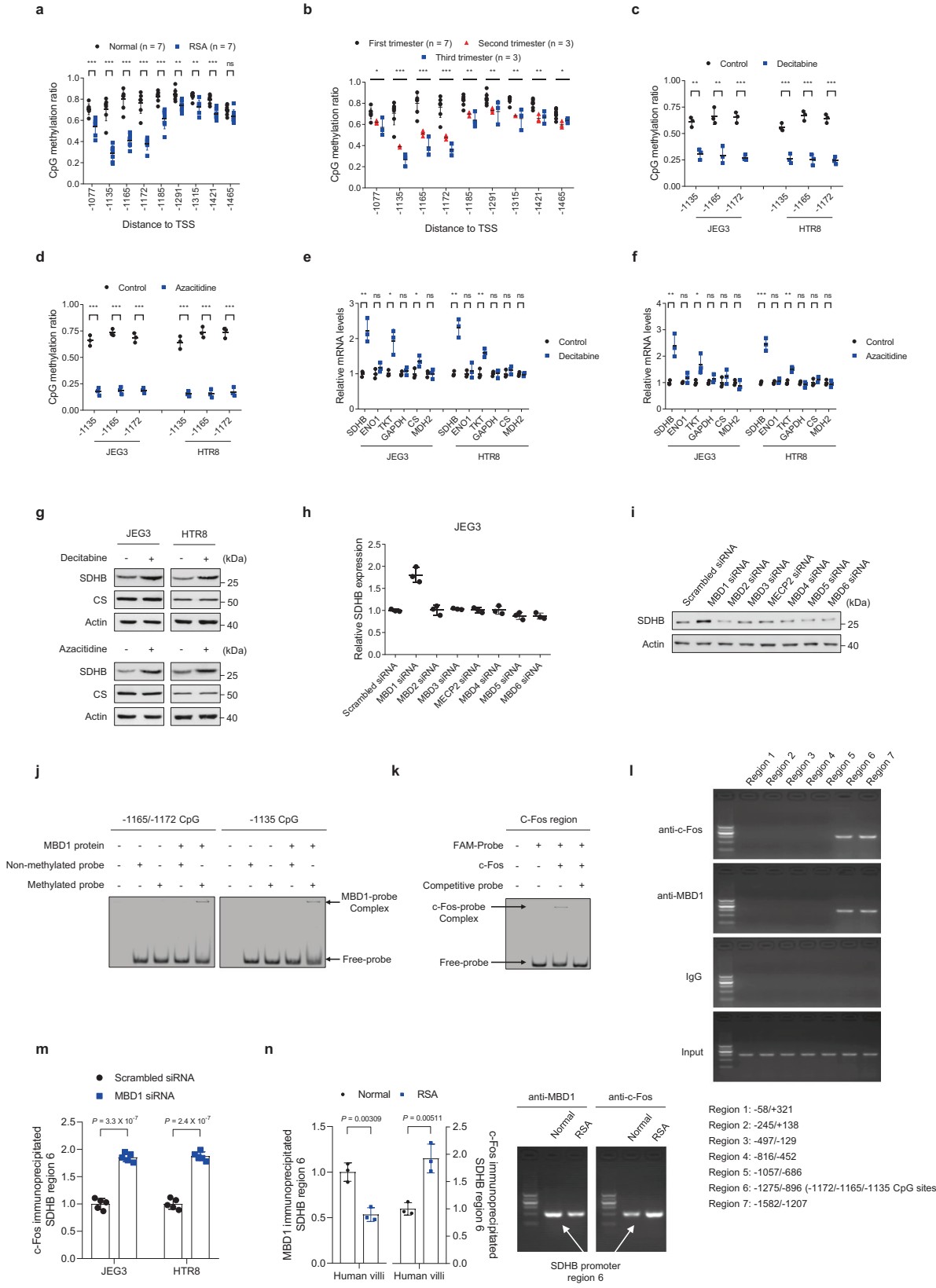

**Succinate accumulation increases HIF-1α levels by mimicking hypoxia in trophoblasts**. Next, we investigated the role of succinate in the invasion and proliferation of extravillous trophoblasts at the maternal–fetal interface. Succinate is known to inhibit prolyl hydroxylase (PHD) in the cytosol, leading to HIF-1α stabilization; therefore, succinate can increase the expression of genes that facilitate angiogenesis, metastasis, and glycolysis, and may facilitate trophoblast adhesion and invasion at the maternal–fetal interface.[22,23] Dimethyl succinate (DMS), which can increase intracellular succinate levels,[24] was found to dose-dependently increase HIF-1α expression in both JEG3 and HTR8 cells alongside the expression of its target genes IL-1β and CXCR4

**Fig. 3 DNA methylation regulates SDHB expression. a** CpG methylation levels in the SDHB promoter region of villi from individuals with RSA and normal controls ($n = 7$ persons/group). Exact $P$ values in **a** are listed below: [−1077] 0.000624, [−1135] $2.5 \times 10^{-6}$, [−1165] $7.6 \times 10^{-7}$, [−1172] $9.9 \times 10^{-7}$, [−1185] 0.000193, [−1291] 0.00272, [−1315] 0.00234, [−1421] $9.6 \times 10^{-5}$, and [−1465] 0.0625. Other CpG sites besides those presented here are shown in Supplementary Fig. 2c. **b** SDHB promoter CpG methylation levels in villi from normal controls in different trimesters ($n = 7$ persons of the first trimester; $n = 3$ persons of the second trimester; $n = 3$ persons of the third trimester). Exact $P$ values in **b** are listed below: [−1077] 0.00489, [−1135] $2.2 \times 10^{-5}$, [−1165] $2.7 \times 10^{-5}$, [−1172] $2.2 \times 10^{-5}$, [−1185] 0.000316, [−1291] 0.00575, [−1315] 0.00011, [−1421] 0.000128 and [−1465] 0.00521. Other CpG sites besides those presented here are shown in Supplementary Fig. 2d. **c, d** CpG methylation levels in JEG3 and HTR8 cells treated with decitabine (**c**) ($n = 3$ independent experiments) or azacitidine (**d**) ($n = 3$ independent experiments). Exact $P$ values in **c** are listed below: JEG3: [−1135] 0.00158, [−1165] 0.00463, and [−1172] 0.000308; HTR8: [−1135] 0.000877, [−1165] 0.000448, and [−1172] 0.000206. Exact $P$ values in **d** are listed below: JEG3: [−1135] 0.000164, [−1165] $3.8 \times 10^{-5}$, and [−1172] $7.4 \times 10^{-5}$; HTR8: [−1135] 0.000202, [−1165] $1.0 \times 10^{-5}$, and [−1172] 0.000112. **e, f** Gene expression in JEG3 and HTR8 cells treated with decitabine (**e**) ($n = 3$ independent experiments) or azacitidine (**f**) ($n = 3$ independent experiments). Exact $P$ values in **e** are listed below: JEG3: SDHB 0.00377, ENO1 0.181, TKT 0.0112, GAPDH 0.339, CS 0.0436, and MDH2 0.973; HTR8: SDHB 0.00112, ENO1 0.559, TKT 0.00433, GAPDH 0.933, CS 0.283, and MDH2 0.763. Exact $P$ values in **f** are listed below: JEG3: SDHB 0.000527, ENO1 0.171, TKT 0.0362, GAPDH 0.204, CS 0.303, and MDH2 0.398; HTR8: SDHB 0.000527, ENO1 0.383, TKT 0.00184, GAPDH 0.349, CS 0.271, and MDH2 0.638. **g** SDHB protein levels in cells treated with decitabine or azacitidine. **h, i** SDHB mRNA (**h**) ($n = 3$ independent experiments) and protein levels (**i**) in cells with knockdown of each methyl-CpG-binding protein. The knockdown efficiencies for each siRNA are shown in Supplementary Fig. 3b, c. **j** Binding abilities between MBD1 and double-strand DNA in the −1165/−1172 or −1135 CpG sites in EMSA. **k** Binding ability between c-Fos and double-strand DNA in the −1165/−1135 CpG sites in EMSA. **l** ChIP followed by PCR showing c-Fos and MBD1 occupancy at the SDHB promoter in cultured JEG3 cells. **m** ChIP-qPCR shows c-Fos occupancy at the SDHB promoter in MBD1-knockdown cells and control cells ($n = 5$ independent experiments). **n** ChIP followed by PCR/qPCR shows c-Fos and MBD1 occupancy at the SDHB promoter in villi from individuals with RSA ($n = 3$ biologically independent samples) and normal controls ($n = 3$ biologically independent samples). The left panel shows the ChIP-qPCR results. The right panel shows a representative result of ChIP-PCR. **a–f, h, m, n** Data represent the mean ± standard error, *$P < 0.05$, **$P < 0.01$, ***$P < 0.001$, ns not significant. Two-tailed unpaired Student's $t$ tests in **a, c–f, h, m, n**; one-way ANOVA in **b**. Source data are provided as a Source Data file.

(Fig. 4a), confirming that HIF-1α was activated. Besides, succinate is known to inhibit the α-KG-dependent TET family of 5-methylcytosine (5mC) hydroxylases, through competing with α-KG.[25] In both JEG3 and HTR8 cells, we demonstrated that although ectopic expression of either TET1 or TET2, resulted in high levels of 5hmC in both JEG3 and HTR8 cells, additional administration of DMS caused a substantial decrease in TET1/TET2-mediated 5hmC production (Fig. 4b). These results validated that succinate competes with α-KG and leads to TET inhibition. Moreover, we found that the DNA methylation levels of SDHB promoter increased, whereas SDHB expression levels decreased in DMS-treated JEG3 and HTR8 cells (Fig. 4c, d). These results suggested the presence of a positive feedback regulation between SDHB methylation and succinate.

HIF-1α is known to promote cell invasion and proliferation;[17,26] therefore, we confirmed that increased DMS levels promoted invasion (Fig. 4e) and proliferation (Fig. 4f) in JEG3 and HTR8 cells. SDHB knockdown also increased the expression of HIF-1α and its downstream targets IL-1β and CXCR4 (Fig. 4g) to promote both JEG3 and HTR8 cell invasion (Fig. 4h) and proliferation (Fig. 4i).

During the early stages of pregnancy, HIF-1α expression in trophoblasts is determined by the balance between oxygen abundance and α-KG-dependent prolyl hydroxylase (PHD). Therefore, increased succinate accumulation induced by reduced SDHB expression could inhibit PHD2 and decrease HIF-1α hydroxylation, blocking its degradation via the Von Hippel–Lindau (VHL) pathway. First, we confirmed that hypoxia induced the expression of HIF-1α and its downstream targets in JEG3 and HTR8 cells; however, this effect was partially blocked by SDHB overexpression (Fig. 4j). In addition, the high-succinate levels induced by SDHB knockdown caused HIF-1α accumulation, even under normoxia, which could be blocked by dimethyl ketoglutarate (DMKG) supplementation in the cultured cells (Fig. 4k). Conversely, HIF-1α accumulation was blocked by SDHB overexpression and DMKG supplementation under hypoxia (Fig. 4l), while the regulatory effects of SDHB overexpression and knockdown on HIF-1α were blocked in both JEG3 and HTR8 cells by either PHD2 or VHL knockdown (Fig. 4m−p). Combined with the finding that DMS increases HIF-1α pathway

signaling even under normoxia (Fig. 4q), these results indicate that succinate provides a hypoxia-like environment for cells. During early pregnancy, succinate accumulates in trophoblasts to produce hypoxia-like cellular conditions; however, defective SDHB promoter methylation (Fig. 3a) can increase SDHB expression (Fig. 1a, b), decrease succinate accumulation (see Table 1), and reverse hypoxia, thus decreasing HIF-1α expression and increasing the risk of RSA (Fig. 4r, s).

**Decreased succinate accumulation contributes toward the onset of abortion in mice.** D-aminolevulinate synthase 1 can catalyze the synthesis of 5-aminolevulinate from succinyl-CoA and glycine to initiate the heme biosynthesis pathway; therefore, glycine supplementation may reduce succinate levels. We confirmed that adding glycine to the culture media decreased succinate levels in both JEG3 and HTR8 cells (Fig. 5a). Intraperitoneal glycine injection (1500 mg/kg/day) led to decreased succinate levels in the villi of mice (Fig. 5b). To investigate whether decreased succinate accumulation impaired embryo implantation and induced spontaneous abortion, we examined embryo implantation efficiency after decreasing succinate levels via intraperitoneal glycine injection. We found that intraperitoneal glycine injection increased the embryo-resorption rate (Fig. 5c) and could be rescued by increasing succinate levels via intraperitoneal DMS injection (Fig. 5b, c). Since SDHB levels did not differ between the different treatment groups (Fig. 5d), these results indicate that succinate levels play a crucial role in embryo implantation and RSA occurrence. Moreover, in the CBA/J × DBA2 spontaneous abortion mouse model, increasing pregnant mice succinate levels via intraperitoneal DMS injection resulted in increased IL-1β levels in mouse villi samples (Fig. 5e), and significantly decreased the embryo-resorption rate (Fig. 5f). Taken together, these findings indicate that succinate can promote the invasion of villi and embryo implantation, with low-succinate accumulation in early pregnancy, increasing the risk of RSA (Fig. 5g).

**Discussion**
Genetic abnormalities, structural abnormalities, infection, endocrine abnormalities, and immune dysfunction have all been

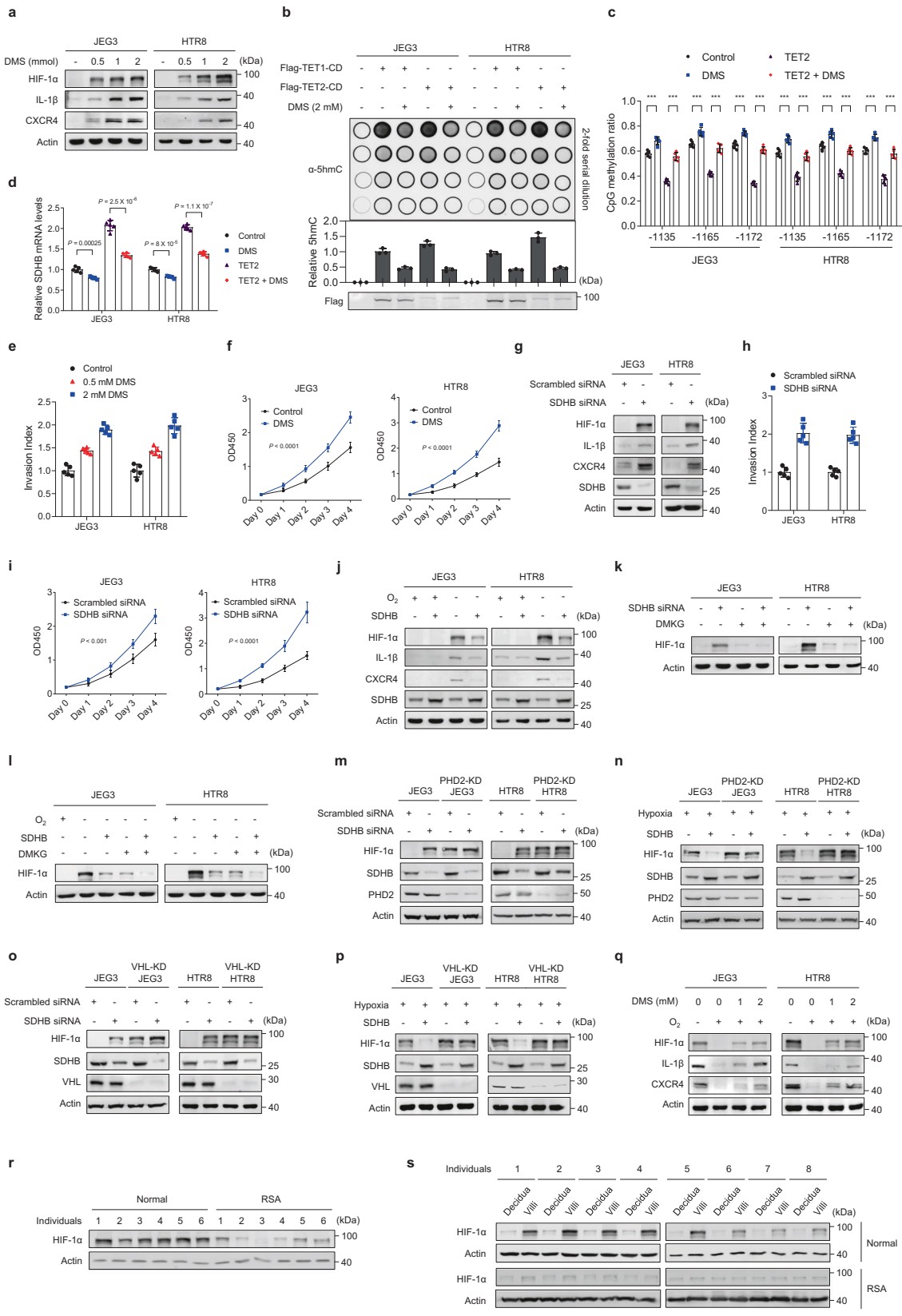

shown to affect the risk of RSA[27] alongside environmental factors such as smoking and cocaine, alcohol, and caffeine consumption;[28] however, the mechanisms underlying RSA occurrence remain elusive. In this study, we investigated the role of metabolites in RSA, finding that high SDHB expression decreased succinate accumulation and induced RSA. Our findings

suggest that the maintenance of high-succinate levels in villi during the first trimester is important for embryo implantation and that low-succinate accumulation may contribute toward the onset of RSA. In addition, we confirmed that the dynamic changes in SDHB expression under physiological conditions and pathological increases in SDHB expression are caused by *SDHB*

**Fig. 4 Succinate increases HIF-1α by mimicking cellular hypoxia. a** Western blot analysis of HIF-1α and its downstream targets in cells treated with different dimethyl succinate (DMS) concentrations. **b** Genomic DNAs were isolated from the cultured cells with various treatments, spotted on nitrocellulose membranes, and immunoblotted with an antibody specific to 5hmC. Quantification of 5hmC was calculated from three independent assays. **c** CpG methylation levels of *SDHB* promoter in JEG3 and HTR8 cells with different treatments (*n* = 5 biological repeats). **d** SDHB mRNA levels in JEG3 and HTR8 cells with different treatments (*n* = 5 biological repeats). **e, f** Invasion (**e**) and proliferation (**f**) of JEG3 and HTR8 cells treated with different DMS concentrations (*n* = 5 biological repeats). **g** Western blot analysis of HIF-1α and its downstream targets in SDHB-knockdown cells and control cells. **h, i** Invasion (**h**) and proliferation (**i**) of SDHB-knockdown cells and control cells (*n* = 5 biological repeats). **j–q** Western blot analysis of HIF-1α and its downstream targets in cells with various treatments. **r, s** Western blot analysis of HIF-1α in tissues from individuals with RSA and normal controls. **c–f, i** Data represent the mean ± standard error, *P < 0.05, **P < 0.01, ***P < 0.001, ns not significant. One-way ANOVA in **c–f, i**. Source data are provided as a Source Data file.

promoter DNA methylation. Interestingly, we confirmed that elevated succinate levels in turn increased the DNA methylation of *SDHB* and decreased the expression of *SDHB*, by inhibiting the activities of the α-KG-dependent TET family. These results suggested the existence of a positive feedback regulation between *SDHB* methylation and succinate. However, which of the succinate-related metabolic element and *SDHB* methylation-related epigenetic element is the initiation factor in normal pregnancy remains unknown. The uncovered causality between succinate levels and *SDHB* methylation requires further investigation. Besides, a SDHB overexpression mouse model is needed to provide in vivo evidence that low succinate induces embryo resorption in future studies.

Maternal hypoxia elicits changes at the maternal–fetal interface via oxygen tension, which may regulate trophoblast differentiation and invasion. Oxygen signals are sensed and transmitted via HIFs by prolyl hydroxylase family proteins and the metabolite α-KG; however, succinate can inhibit prolyl hydroxylase activity as it is a competitive inhibitor of multiple α-KG-dependent dioxygenases.[22,23,29] Here, we found that succinate maintained hypoxia in extravillous trophoblasts from human tissue and cultured cells and stabilized HIF-1α via the PHD2-VHL pathway. HIFs are a family of basic helix–loop–helix transcription factors that mediate the response to changes in cellular oxygen levels.[30] During the first trimester, HIF-1α and -2α, which are expressed in the human placenta, are localized in the syncytiotrophoblast (STB), cytotrophoblasts, and fetoplacental vascular endothelium,[31] with their mRNA and protein levels peaking at 7–10 weeks of gestation and declining thereafter.[32] In this study, SDHB expression decreased and succinate levels increased at 7–10 weeks of gestation and declined thereafter, indicating a strong correlation between succinate and HIFs, which play key roles in regulating extravillous trophoblast invasion. For instance, trophoblast invasion is 17% lower in Hif-1α and Hif-2α double-knockout mice, which display impaired placental vascularization.[17] IGF-II is positively regulated by HIFs and induces extravillous cytotrophoblast migration in vitro,[33] although there is currently no direct evidence of its regulation by HIFs in the placenta. Succinate can also act as an inflammatory signal via HIF-1α to induce IL-1β,[34] which can directly promote the motility of first-trimester extravillous trophoblasts.[35] We found although the CBA/J × DBA/2 spontaneous abortion mouse model was not accompanied by SDHB overexpression and succinate deficiency in villi tissue (Supplementary Fig. 4a, b), additional supplementation of DMS further increased IL-1β level in mice villi tissue and decreased the embryo resorption in this spontaneous abortion mouse model. Although we could not obtain the direct evidence that increased succinate accompanied the increased trophoblast invasion/proliferation during the first trimester of pregnancy due to technical limitations with sample collection, these findings emphasize the importance of succinate accumulation during the first trimester, given that low-succinate accumulation could reduce extravillous trophoblast invasion and proliferation and increase the risk of

RSA. Further, these results at least partially explained why the invasive/proliferative properties of human extravillous trophoblasts increase during the first trimester and decrease following embryo implantation under physiological conditions.

Our data suggest that monitoring villous succinate levels could help predict recurrent abortion in RSA patients, and intravenously increasing succinate levels during the first trimester could reduce the risk of RSA. Other strategies for increasing succinate accumulation should be investigated in future studies; for example, aspirin can specifically bind to SDHA and SDHC, but not SDHB and SDHD (Supplementary Fig. 5a), and reduce its activity in an uncompetitive manner (Ki = 1.19 mM; Supplementary Fig. 5b, c) while increasing succinate levels in cultured cells (Supplementary Fig. 5d). Although high aspirin concentrations elicit many side effects in humans, further studies should explore its ability to inhibit SDH and increase succinate accumulation to prevent and treat RSA.

## Methods

**Patients.** Human samples from RSA patients (*n* = 30) and normal pregnancy (*n* = 30) were obtained from Shanghai First Maternity and Infant Hospital between June 1, 2016 and Aug 31, 2020. The sample size is calculated by a two-sample T-test power analysis to meet the threshold value of power (0.95) to tell the significant difference of metabolite concentration between the case group and the control group. Study design and conduct were approved and supervised by the Ethics Committee of Shanghai First Maternity and Infant Hospital (https://www.51mch.com/news/content/id/287/pid/18552) through Ethics Vote KS18133 in accordance with the criteria set by the Declaration of Helsinki. The Ethics Committee of Shanghai First Maternity and Infant Hospital can be contacted by writing to the Office of Ethics Committee, Shanghai First Maternity and Infant Hospital, 2699 West Gaoke Road, Pudong District, 201300 Shanghai or sending an email to shsdyfybjyyxllwyh@126.com or by calling +86-021-20261211. All patients with normal pregnancy and patients with RSA who underwent an induced abortion in Shanghai First Maternity and Infant Hospital provided written informed consent. This case–control study was reported to follow the STROBE guidelines[36] (STROBE checklist is provided in Supplementary Table 4).

**Mice.** Eight-week-old C57BL/6 mice were purchased from Shanghai SLAC Laboratory Animal Company Limited and housed in the animal care facility of Anhui Medical University under standard pathogen-free conditions with a 12-h light/dark schedule and provided with food and water ad libitum, the temperature was between 20 and 24 °C and relative humidity between 45 and 65 rH. All the studies with research animals complied with the Experimental Animal Management Regulations of China. All experiments were reviewed and approved by the Institutional Animal Care and Use Committee of the institute at Anhui Medical University.

**Tissue collection.** First-trimester villous and decidual tissues were obtained from the placentas of individuals with clinically normal pregnancies (age ranged from 22 to 35, 28 ± 3 years; gestational age at sampling, 52 ± 9 days, mean ± SD) that had been terminated for nonmedical reasons. Second-trimester placentas were obtained from individuals with clinically normal pregnancies (age ranged from 22 to 34, 28 ± 3 years) that had been terminated due to unplanned pregnancy or family planning. Third-trimester placentas were obtained from individuals (age ranged from 24 to 33, 28 ± 3 years) who had undergone normal vaginal delivery. Villous samples were also obtained from patients with RSA (age ranged from 22 to 35, 28 ± 4 years, gestational age at abortion, 53 ± 7 days) who had experienced more than three unexplained and consecutive spontaneous abortions. The gestational age of RSA patients and first-trimester normal pregnancies were paired. Among these patients, we excluded cases with co-existing health problems, including (1) infection, by

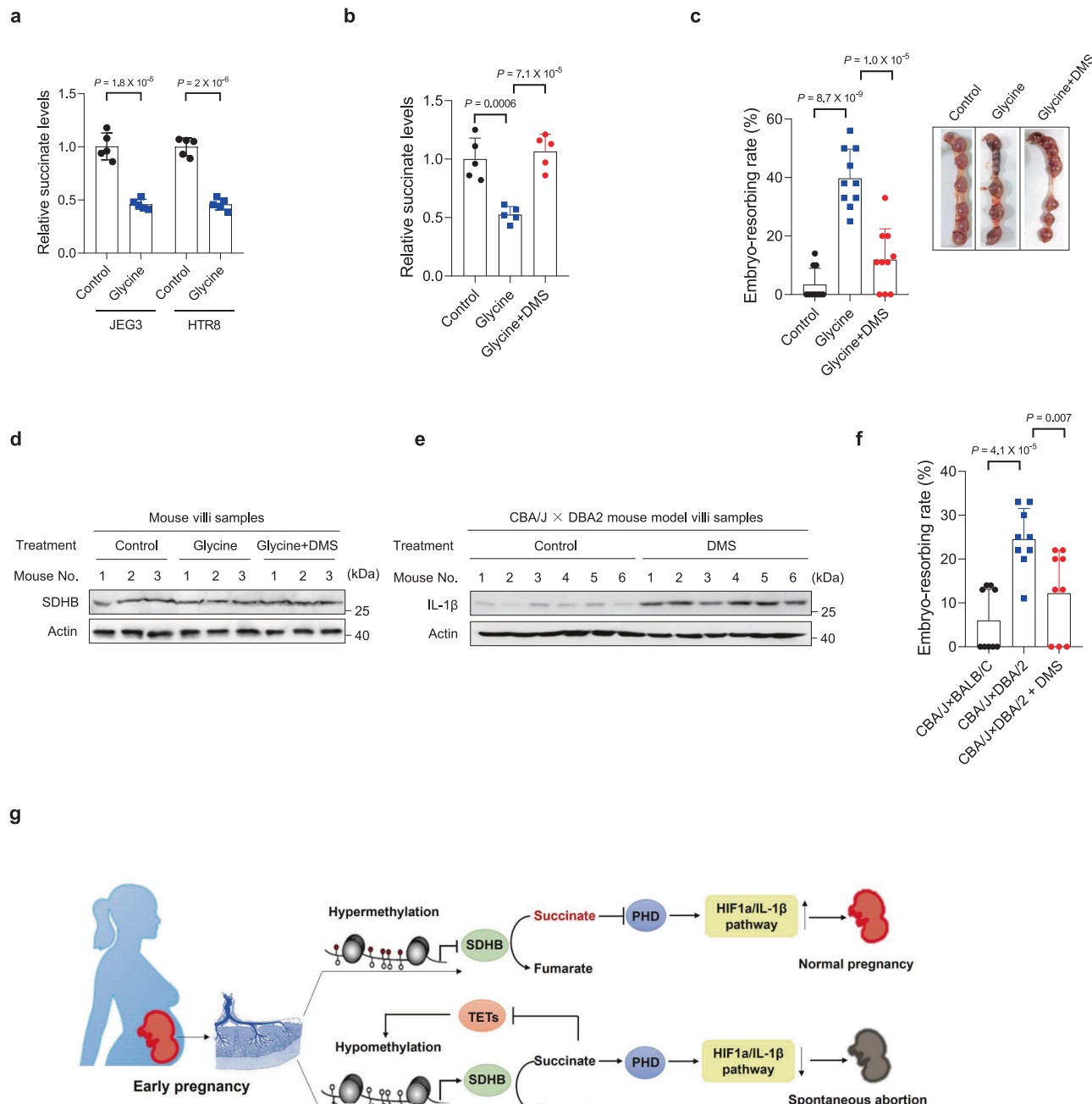

**Fig. 5 Decreased succinate accumulation contributes toward the onset of abortion in mice. a** Succinate levels in cells treated with or without glycine ($n = 5$ biological repeats). **b** Succinate levels in villi of mice with various treatments ($n = 5$ biological repeats). **c** Embryo-resorption rates in mouse models with various treatments ($n = 10$ biological repeats). Representative macroscopic appearance of uteri on gestational day 13.5 in pregnancies with various treatments were shown in right panel. **d** SDHB expression levels in villi from mice with various treatments. **e** IL-1β expression levels in villi from mice after indicated treatments. **f** Embryo-resorption rates in mouse models with indicated treatments ($n = 9$ biological repeats). **g** Schematic illustration indicating how low embryonic villous succinate accumulation increases RSA risk. **a–c, f** Data represent the mean ± standard error, $*P < 0.05$, $**P < 0.01$, $***P < 0.001$, ns not significant. Two-tailed unpaired Student's $t$ tests in **a**; one-way ANOVA in **b, c, f**. Source data are provided as a Source Data file.

taking body temperature and blood examination, (2) endocrine or metabolic diseases, including polycystic ovarian syndrome, hyperprolactinemia, hypothyroidism, hyperthyroidism, diabetes mellitus, etc., by testing hormone levels, thyroid function, and blood glucose levels, (3) chromosomal abnormalities, by conducting villi low-density chip to exclude chromosome number abnormalities, (4) anatomical abnormalities, including congenital malformation of uterus, cervical incompetence, intrauterine adhesions, uterine fibroids, adenomyosis, etc., (5) autoimmune diseases, including antiphospholipid antibody syndrome, systemic lupus erythematosus, Sjogren's syndrome, etc., by testing immune index. The above criteria were designed to exclude known causes or risk factors of RSA. However, there still might be some unknown confounding factors that may lead to unaddressed confounding bias. After curettage, the tissues were immediately collected under sterile conditions into pre-chilled PBS and divided into villi and decidua. Then the villus and decidual tissues were washed again with pre-chilled PBS to exclude contamination of the villi by the decidua and vice versa. The samples were dried with paper towels and cut into small pieces. For metabolites analysis, samples were fixed by adding five volumes of pre-chilled 70% methanol and stored at −80 °C until use. For western blotting, mRNA extraction, or DNA extraction, samples were frozen in liquid nitrogen and stored in the refrigerator at −80 °C until use. All sample collection and utilization procedures were approved by the Human Research Ethics Committee of Shanghai First Maternity and Infant Hospital. All subjects provided informed consent for tissue sample collection.

**SDHB promoter region sequencing**. The villi of 10 normal pregnant women and 20 RSA patients were collected. After genome extraction, the *SDHB* promoter region (−1500~+200) was sequenced by Sanger sequencing. The primer sequences are shown in Supplementary Table 3.

**Cell culture and treatment**. The human choriocarcinoma cell lines JEG3 and HTR8 were obtained from the American Type Culture Collection (ATCC) and authenticated by DNA fingerprinting with small tandem repeat (STR) profiling. The JEG3 and HTR8 cells were cultured in Dulbecco's Modified Eagle's Medium (DMEM) (HyClone, Pittsburgh, VA, USA) supplemented with 10% fetal bovine serum (FBS; Biochrom, Cambridge, UK), 100 units/mL of penicillin, and streptomycin (Invitrogen, Waltham, MA, USA). Cells were transfected with plasmids using polyethylenimine (PEI, linear, 25 KDa) or treated with drugs by adding glycine (5 mM, final concentration), dimethyl succinate, and dimethyl 2-oxoglutarate (1 mM, final concentration) to the culture medium 8, 6, and 4 h before harvesting, respectively. Hypoxia was induced by adding EC-Oxyrase (0.3 Unit/mL, final concentration) to the culture medium 6 h before harvesting.

**Plasmid construction**. SDHB was cloned into the pcDNA3.1b-C-Flag plasmid with a Flag tag at the C-terminus of the SDHB gene. The primers and restriction enzyme cutting sites used for plasmid construction are shown in Supplementary Table 3. The constructions of TET1 and TET2 plasmids were described in our previous study.[37]

**RNA interference**. Small RNA interference (siRNA) and stable knockdown (shRNA) were used in this study. Double-stranded human SDHB, MBD1-6, and MECP2 gene siRNAs were purchased from GenePharma (ShangHai, China) and transfected into cells using RNAiMax (Invitrogen) according to the manufacturer's instructions. The sequences of siRNA targets used in this study were listed in Supplementary Table 3.

Stable shRNA knockdown cells were produced by co-transfecting cells with pCMV-VSV-G, pCMV-Gag-Pol, and shRNA plasmids using the calcium phosphate method. The transfected cells were cultured in DMEM containing 10% FBS for 24 h. After 24 h, the supernatant of the culture medium was collected and used as a retroviral preparation to infect cells at 10% confluency in 90-mm culture dishes. Cells were re-infected 48 h after the initial infection, and 5 μg/mL of puromycin (Invitrogen) was used to select infected cells. The sequences of shRNA targets are listed in Supplementary Table 3. Knockdown efficiency was verified by qPCR or western blotting.

**Quantitative real-time PCR**. RNA from cultured cells or human tissue samples was prepared with TransZol (Trans Gen Biotech), and cDNA was synthesized from 5 μg of RNA using TransScript First-Strand cDNA synthesis SuperMix (Trans Gen Biotech). Gene expression was determined by real-time PCR using an iQTM SYBR Green SuperMix Kit (Bio-Rad, CA, USA) with a CFX96TM Real-Time system (Bio-Rad). All data were normalized to ACTB expression. The primer sequences are listed in Supplementary Table 3.

**Measurements of mitochondrial levels**. We used the ratio of the mitochondrial DNA to nuclear DNA (mtDNA/nDNA) to indicate the mitochondrial levels. The primers used for quantifying mitochondrial DNA (16 S rRNA) and nuclear DNA (β2-microglobulin) are shown in Supplementary Table 3.

**NMR measurements**. Villi or decidua tissues (about 50 mg) were extracted with 600 μL of precooled methanol–water mixture (2/1, v/v) using a tissue lyser (QIAGEN TissueLyser II, Germany). Supernatant for each sample was collected respectively after 10 min of centrifugation (11,180 × g, 4 °C). Such extracting procedure was further repeated twice. Three supernatants obtained for each sample were combined and centrifuged (16,099 × g, 4 °C) for another 10 min. The resultant supernatant for each sample was respectively lyophilized after removal of methanol in vacuo. The dried extracts were reconstituted into 600 μL of phosphate buffer (0.15 M, $K_2HPO_4$–$NaH_2PO_4$, pH 7.43) containing 80% $D_2O$ (v/v) and trimethylsilyl propionate (TSP, 0.2915 mM). The mixture was then centrifuged at 16,099 × g for 10 min at 4 °C. Then, 550 μL each supernatant was transferred into a standard 5-mm NMR tube for analysis.

All the one-dimensional $^1H$ NMR spectra were acquired at 298 K on a Bruker Advance III 600 MHz NMR spectrometer (600.13 MHz for proton frequency) equipped with a quaternary cryogenic inverse probe (Bruker Biospin, Germany) using the first increment of the gradient selected NOESY pulse sequence (NOESYGPPR1D). In total, 64 transients were collected into 32-k data points with a spectral width of 20 ppm for each sample. The total relaxation delay time was 26 s, which allowed the completely relaxed NMR spectra to be obtained.

For resonance assignment purposes, a series of two-dimensional (2D) NMR spectra were acquired for selected samples and processed as previously reported.[38–40] These included $^1H − ^1H$ correlation spectroscopy (COSY), $^1H − ^1H$ total correlation spectroscopy (TOCSY), J-resolved spectroscopy (JRES), $^1H − ^{13}C$ heteronuclear single quantum correlation (HSQC), and $^1H − ^{13}C$ heteronuclear multiple bond correlation (HMBC) 2D NMR spectra.

**Quantification of metabolites**. All the NMR spectra were processed using the software package TOPSPIN (V3.6.0, Bruker Biospin, Germany). For $^1H$ NMR spectra, an exponential window function was employed with a line broadening factor of 1 Hz and zero-filled to 128 k prior to Fourier transformation. Each spectrum was then phase- and baseline-corrected manually with the chemical shift referenced to TSP (δ 0.00). The spectral regions were then integrated into bins with a width of 0.002 ppm (1.2 Hz) using AMIX software package (V3.8.3, Bruker Biospin). The absolute concentration of metabolites was calculated with the known concentration of TSP.

In an incompletely relaxed NMR spectra obtained in our experiments, the integral area for a given proton resonance obeys the following equation (Eq. 1):

$$A^m = A_0^m[1 - \exp(-t/T_1^m)] \tag{1}$$

where $t$ is total relaxation time (i.e., RD plus acquisition time); $A_0^m$ and $A^m$ are the integral areas for the signals of a given proton $m$ (with a spin-lattice relaxation time of $T_1^m$) in the completely relaxed state and a given spectrum, respectively. The signal integrals for proton m and internal reference TSP in a given spectrum have the following relationship (Eq. 2):

$$\frac{A_0^m}{A_0^{TSP}} = \frac{A^m}{A^{TSP}} * \frac{1 - \exp(-t/T_1^{TSP})}{1 - \exp(-t/T_1^m)} \tag{2}$$

where $A_0^{TSP}$ and $A^{TSP}$ are the integrals for methyl groups of TSP in the completely relaxed and a given spectrum, respectively; $T_1^{TSP}$ is the spin-lattice relaxation time for methyl protons of TSP. The concentration of a metabolite ($C_m$) and TSP ($C_{TSP}$) in the same spectra obeys the relationship described in Eq. 3 taking into consideration of proton numbers related to the NMR signals (Eq. 3), where $N_m$ and $N_{TSP}$ denote proton numbers for the corresponding metabolite signal and the methyl groups of TSP (i.e., nine protons), respectively. Using Eq. 3, the concentration of a given metabolite can be calculated with the known concentration of TSP.

$$C_m = \frac{N_{TSP}C_{TSP}}{N_m} * \frac{A_0^m}{A_0^{TSP}} = \frac{N_{TSP}C_{TSP}}{N_m} * \frac{A^m}{A^{TSP}} * \frac{1 - \exp(-t/T_1^{TSP})}{1 - \exp(-t/T_1^m)} \tag{3}$$

**DNA methylation analysis**. DNA methylation at specific CpG sites was determined by MethylTarget sequencing (Genesky Biotechnologies Inc., Shanghai, China), using next-generation sequencing-based multiple targeted CpG methylation analysis.[41,42] Primer design and validation were performed using Methylation Primer software on bisulfate-converted DNA. Primer sets were designed flanking each targeted CpG site in a 100–300 nucleotide region and are shown in Supplementary Table 3. Genomic DNA was extracted from frozen samples using Genomic Tip-500 columns (Qiagen, Valencia, CA, USA) and from bisulfite-converted samples using an EZ DNA Methylation™-GOLD Kit (Zymo Research, CA, USA) according to the manufacturer's instructions. After PCR amplification (HotStarTaq polymerase kit, TAKARA, Tokyo, Japan) and library construction, samples were sequenced (Illumina HiSeq Benchtop Sequencer, CA, USA) by paired-end sequencing according to the manufacturer's guidelines.[43]

**Dot-blot assays**. For dot-blot assays, we followed the procedures described previously.[37] Briefly, genomic DNA was spotted on nitrocellulose membranes. The membrane was baked at 80 °C and then blocked with 5% skimmed milk in TBST for 1 h, followed by the incubation with the anti-5-hmC antibody overnight at 4 °C and HRP-conjugated anti-rabbit IgG secondary antibody for 1 h at room temperature. After washing three times with TBST, the membrane was treated with ECL and scanned by a Typhoon scanner. The quantification of dot-blot was done by Image-Quanta software (GE Healthcare).

**Cell invasion assay**. The invasion of JEG3 and HTR8 cells through Matrigel was evaluated objectively by counting the number of the cells that transferred through the membrane in the invasion chamber. The upper surface of the filter (8-μm pore size, 6.5 mm diameter, PET membrane; Corning Life Sciences, Corning, NY, USA) in a Transwell plate (24-well plate, Corning, NY, USA) was pre-coated with 15 μL of Matrigel (BD Biosciences) and air-dried under sterile conditions. Prior to use, the Matrigel was rehydrated with 100 μL of warm DMEM/F12 (Gibco; Thermo Fisher Scientific, Inc.) for 2 h. JEG3 cells ($1 \times 10^5$) in 100 μL of serum-free DMEM/F12 were seeded into the upper part of the chamber, while 600 μL of DMEM/F12 with 10% FBS was added to the lower chamber. To study the effect of succinate on invasion, cultured JEG3 and HTR8 cells were treated with 0.5 mM or 2 mM dimethyl succinate for 2 h prior to invasion assays. After seeding the cells in the upper part of the chamber, we added 0.5 mM and 2 mM dimethyl succinate in the medium of the lower chamber, respectively. To study the role of SDHB in invasion, either SDHB-knockdown or wild-type cells were seeded into the upper part of the chamber. The cells were then incubated for 16 h in 5% $CO_2$ at 37 °C, after which the cells attached to the upper surface of the filter were removed by scrubbing with a cotton swab. The cells remaining on the lower surface were fixed in methanol for 15 min at 15 °C, stained with DAPI for 5 min, and quantified by counting the cells that had migrated to the lower surface under a fluorescent Inverted microscope in five random fields. Each experiment was carried out in three replicates.

**Cell proliferation assay**. For DMS treatment, JEG3 and HTR8 cells were plated and incubated (37 °C, 5% CO$_2$) in 96-well plates (2 × 10$^3$ cells/well) and treated with 2 mM DMS for 24~96 hours. The cultured medium was changed daily until the cells were harvested, and then incubated with CCK-8 solution (C0039, Beyotime Biotechnology) for 2 h, and their absorbance measured at 450 nm. For SDHB siRNA interference, scrambled siRNA or SDHB siRNA were transfected into JEG3 or HTR8 cells using RNAimax (Invitrogen). Transfected cells were then counted and plated and incubated in 96-well plates (2 × 10$^3$ cells/well) for 24–96 h followed by CCK-8 solution (C0039, Beyotime Biotechnology) treatment, and their absorbance was measured at 450 nm.

**Western blot analysis**. Cells were lysed in EBC lysis buffer (50 mM Tris HCl pH 8.0, 120 mM NaCl, 0.5% Nonidet P-40) supplemented with protease and phosphatase inhibitors (Selleck Chemicals). Proteins were separated by 10% SDS-PAGE gel and blotted with indicated primary antibodies. The following primary antibodies were used for western blot analysis: anti-HK1 (1:1000; #2024; CST), anti-GPI (1:1000; #57893S; CST), anti-TPI (1:500; 10713-1-AP; Proteintech), anti-PGK1 (1:2000; 17811-1-AP; Proteintech), anti-ENO1 (1:500; 11204-1-AP; Proteintech), anti-PKM2 (1:2000; #4053; CST), anti-6PGD (1:1500; #13389S; CST), anti-TKT (1:2000; 11039-1-AP; Proteintech), anti-PDHA1 (1:2000; #3205; CST), anti-CS (1:500; #14309; CST), anti-IDH1 (1:1000; #66969; CST), anti-IDH2 (1:1000; #56439; CST), anti-IDH3A (1:1000; 15909-1-AP; Proteintech), anti-OGDH (1:1000; #26865; CST), anti-SUCLA2 (1:5000; 12627-1-AP; Proteintech), anti-SDHA (1:2000; 14865-1-AP; Proteintech), anti-SDHB (1:1000; 10620-1-AP; Proteintech), anti-SDHC (1:1000; 14575-1-AP; Proteintech), anti-FH (1:1000; #4567; CST), anti-MDH2 (1:1000; 15462-1-AP; Proteintech), anti-HIF1a (1:500; #36169; CST), anti-IL-1β (1:1000; #12703; CST), anti-CXCR4 (1:500; 11073-2-AP 57893S; Proteintech), anti-VHL (1:1000; #68547S;CST), and anti-PHD2 (1:1000; #4835; CST), anti-MBD1 (1:1000; ab108510; Abcam), anti-MBD2 (1:1000; 55200-1-AP; Proteintech), anti-MBD3 (1:1000; 14258-1-AP; Proteintech), anti-MBD4 (1:1000; 11270-1-AP; Proteintech), anti-MBD5 (1:1000; 15961-1-AP; Proteintech), anti-MBD6 (1:1000; ab204403; Abcam), anti-MECP2 (1:1000; 10861-1-AP; Proteintech), and anti-5-hmC (1:1000; 39769; Active Motif). Western blot gel images were obtained using a Typhoon FLA 9500 scanner (GE Healthcare).

**ATP and SDH activity measurements**. To measure ATP, cells in suspension were mixed with an equal volume of CellTiterGlo in solid white luminescence plates (Grenier Bio-One) according to the manufacturer's instructions (Promega). Luminescence was measured using a GloMax® 96 microplate luminometer (Promega) to obtain ATP activity in relative luciferase units (RLU). To measure cellular SDH activity, treated cells were harvested by trypsinization, washed twice in PBS, and subjected to a Succinate Dehydrogenase Activity Assay Kit (K660-100, Biovision) according to the manufacturer's instructions. The relative SDH activities in tissue were normalized to the citrate synthase activity measured by CS activity kit (K287-100, Biovision) according to the manufacturer's instructions.

**Electrophoretic mobility shift assay (EMSA)**. The c-Fos and MBD1 genes were inserted into pET22b downstream of a sequence encoding hexahistidine and the recombinant protein was purified by affinity chromatography using an Ni+ column. EMSA was performed using an EMSA Kit (Invitrogen) according to the manufacturer's protocols. Labeled single-stranded probes were synthesized by Generay (ShangHai, China), annealed with an annealing reagent (Invitrogen), and incubated with the recombinant protein at 15 °C for 30 min. The mixtures were analyzed using 6% native polyacrylamide gel and FAM-labeled DNA was detected using a Typhoon FLA 9500 scanner (GE Healthcare). The probes used are listed in Supplementary Table 3.

**Chromatin immunoprecipitation (ChIP) assays**. ChIP assays were conducted using an EZ ChIP kit (Upstate). First, cultured cells and villous samples from humans were crosslinked with 1% formaldehyde for 10 min, and DNA was sonicated into fragments with a mean length of 200–500 bp. Sheared chromatin was immunoprecipitated with antibodies against c-Fos, MBD1, or non-specific rabbit IgG (Santa Cruz) overnight at 4 °C and the precipitated DNA fragments were identified by PCR and quantified by real-time qPCR using the primers listed in Supplementary Table 3.

**Drug affinity responsive target stability (DARTS)**. Recombinant Flag-tagged SDHA, SDHB, SDHC, and SDHD proteins were expressed in HEK293T after transient transfection, purified with anti-Flag M2 agarose beads, and eluted with 3 × Flag peptide. After centrifuging for 10 min at 18,000 × g at 4 °C, the supernatants were transferred to a new 1.5-mL tube and added appropriate volume of 10 × TNC buffer to make a final concentration of 1 × TNC buffer. After that, each protein was equally divided and incubated with DMSO or 1 mM aspirin for 1 h followed by a BCA protein quantification to ensure an equal amount of protein per group. Each sample was proteolyzed at room temperature for 30 min with Pronase (1:2000). Lastly, protein samples were separated by SDS-PAGE and analyzed by western blotting.

**Mouse experiments**. For the low-succinate mouse model, 8-week-old female C57BL/6 mice (weight: 20–23 g) were divided into three equal groups using a random number table by body weight, age, and family: control group (vehicle intraperitoneal injection), glycine intraperitoneal injection group (glycine: 1500 mg/kg/day), and glycine + DMS intraperitoneal injection group (glycine: 1500 mg/kg/day + DMS: 3000 mg/kg/day). The day of copulatory plug appearance was arbitrarily designated as day 0.5 of gestation. The intraperitoneal injection was performed from day 0.5–5.5. The embryo absorption rate and implantation number were counted on day 14.5.

For the recurrent abortion mouse model, 8-week-old CBA/J female mice were mated to 8-week-old DBA/2 male mice. As the control, 8-week-old CBA/J female mice were mated to 8-week-old BALB/C male mice as a control mating combination. For each combination, we established two groups: the control group (vehicle intraperitoneal injection), and DMS intraperitoneal injection group (DMS: 3000 mg/kg/day).

The percentage of fetal loss (embryo absorption rate) was calculated as follows: percentage fetal loss $= R/(R + V) \times 100$, where $R$ represents the number of hemorrhagic implantations (sites of fetal loss) and $V$ represents the number of viable surviving fetuses.

**Statistics and reproducibility**. Statistical analysis was performed using Prism 6.0 software (GraphPad Software, Inc.), Excel (Microsoft Corp.), and R version 2.17. Pooled results were expressed as the mean ± SD or SEM. One-way ANOVA was performed for multigroup analyses, while unpaired two-tailed Student's $t$ tests were performed for two-group analyses. Benjamini & Hochberg method was used to correct the $P$ values of $t$ test for metabolites involving multiple sampling. Differences were considered statistically significant if the $P$ value was less than 0.05. Significance is indicated as follows: *$P < 0.05$, **$P < 0.01$, ***$P < 0.001$. For other graphs showing representative data, reproducibility is stated below: $n \geq 3$ biologically independent experiments for Figs. 3g, 3i-l, 4a, 4g, 4j-s, 5d-e and Supplementary Figs. 1c, 3c, 4a, 5a.

**Reporting summary**. Further information on research design is available in the Nature Research Reporting Summary linked to this article.

## Data availability

All raw *SDHB* promoter sequencing and MethylTarget sequencing data created in this study have been uploaded to the Genbank database (https://www.ncbi.nlm.nih.gov/genbank/) with the accession numbers MW922474 to MW922503. All raw *SDHB* promoter MethylTarget sequencing data created in this study have been uploaded to the GEO database (https://www.ncbi.nlm.nih.gov/geo/) with the accession number GSE171877. All the NMR raw data are deposited in the BMRbig database (https://bmrbig.org/) with the accession number bmrbig7. The dataset of patients hospitalized at Shanghai First Maternity and Infant Hospital is stored on a server at Shanghai First Maternity and Infant Hospital. Data generated and/or analyzed in this study, excluding participants' identifying personal information, are available from the corresponding author Jianyuan Zhao (zhaojy@fudan.edu.cn) on reasonable request to protect research participant privacy. Source data are provided with this paper.

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

## Acknowledgements

This work was supported by Grants from the National key research and development program (2019YFA0801900, 2020YFA0803600, 2017YFC1001401, SQ2018YFC100242, 2018YFA0801300, 2018YFC1004700, 2018YFA0800300), and the National Science Foundation of China (Nos. 31330023, 81722021, 81471454, 81730039, 81671460, 31671453, 81771627, 31521003, 81672544, and 81802525).

## Author contributions

J.Y.Z. and L.P.J. conceived the concept, designed and supervised the experiments; X.H. W., S.X., X.Y.Z., R.Z., Y.L., and H.T. performed the experiments; X.H. W. and X.Y.Z. collected the clinic samples; S.X., J.C., W.D.Z., W.X., M.Q.L. and S.M.Z. analyzed the experimental data; J.Y.Z. wrote the manuscript. All authors read and discussed the manuscript.

## Competing interests

The authors declare no competing interests.
