## [Peer Review File · Nature Communications]

Reviewers' Comments:

Reviewer #1:

Remarks to the Author:

In this manuscript the authors demonstrated that the decreased succinate in villi was responsible for increased RSA risk, and DNA methylation in SDHB promoter led to decreased SDHB expression which contributed to the decreased succinate levels. The authors utilized in vitro cells and mice models to reveal the essential roles of succinate in increasing trophoblast invasion/proliferation and preventing pregnancy loss. Overall, this manuscript is well organized. However, in this manuscript, the authors are somewhat exaggerating their findings and additional experiments are needed to confirm the authors' conclusion. Moreover, language editing would be helpful. Specific comments are provided below.

1. In the Summary (Lines 30-31), the authors are overinterpreting their findings about the role of high succinate for embryo implantation. Although there is a correlation between first trimester of pregnancy and high succinate, without evidence showing the increased succinate accompanied with increased trophoblast invasion/proliferation during the first trimester of pregnancy, the conclusion regarding the essential role of high succinate for embryo implantation should be drawn with caution.
2. Line 48, please revise "in the maternal fetal-interface" to "at the maternal-fetal interface". Overall, the language of this manuscript needs to be further refined.
3. Line 71, the role of hypoxia in regulating EVT differentiation has been described previously, thus the authors' statement is inaccurate, especially previous publications have reported the inhibitory effect of hypoxia on EVT invasion.
4. Line 96, the sentence should be "that metabolic alterations in villi is correlated with the occurrence of RSA," rather than "contribute toward"
5. Results: the authors need to state if the clinical samples used for screening from normal pregnancies and RSA patients are paired at gestational age.
6. Line 141, "are required", overexaggerating the findings again.
7. Line 207, "HIF-1a is known to promote cell invasion", reference is required. "
8. Lines 232-246, to directly demonstrate the essential role of succinate in preventing RSA occurrence, the supplementation of succinate to RSA mice models would be helpful. Moreover, instead of using glycine to decrease succinate expression in vivo, SDHB DNA methylation or SDHB overexpression in vivo would be preferred to be carried out followed by embryo-resorption rate and succinate level measurements, which could provide straightforward in vivo evidence .
9. In Discussion, the shortages of this study should be mentioned and discussed.
10. Methods: How long the JEG3 and HTR8 were treated with succinate followed by cell invasion and proliferation assays?
11. Figure 5C: Please provide the representative photo for embryo-resorbing rate.

Reviewer #2:

Remarks to the Author:

Overview

I found this to be a fascinating paper that contained an interesting idea that both impacts on our understanding of the role of metabolite signalling in embryo genesis. As the authors are working with human tissue the experiments that are possible are obviously circumscribed. Furthermore, as with all ideas that are breaking new ground there are many more experiments that could be done. I have a number of technical quibbles, but again these may not be possible to address given the tissue samples used. While I am generally supportive of this paper there are a few points I'd like to see addressed with more experimentation before publication. In addition, there are some interpretational issues that I feel could be clarified.

Major points

1 The role of SDHB and its changes in expression and protein levels in RSA are well supported by the data. However, the relationship between expression of one of four subunits of SDH and its activity is not clear. For the hypothesis being assessed the activity of SDH is key and has to be measured. This should be done on human tissue homogenates and normalised to citrate synthase activity. It would also be useful to compare mtDNA levels to marker nuclear genes. In addition, it is straightforward to assess SDH activity on histological samples. This can also be done relative to cytochrome oxidase. These assays would add a lot and in my view are essential.

2 I have some technical concerns about the way in which the metabolomics was done. For metabolites, most especially succinate, there can be extensive changes in tissues post harvesting. For many of the studies in my lab we clamp freeze the tissue at liquid nitrogen temperature using Wollenberger clamps. Depending on the physiological context, a few seconds where the tissue is not perfused but stays warm can be critical. Now, I recognise that given the constraints in which these samples were analysed this may be challenging. Even so, it is essential that a full description and timings of how the samples were obtained, cooled and stored is given. Limitations to the approach used should be discussed – and the important thing is that succinate seems to change selectively between control and RSA.

A related issue is the way in which the samples are just homogenised in phosphate buffer. Usually in this field frozen samples are first rapidly homogenised in a denaturing solvent – e.g. acid, acetonitrile, methanol etc – to ensure that there is no enzymatic activity during resuspension. Again, it may be challenging to do this. But even so, a control in which mock extractions of ^{13}C -succinate \pm tissues are carried out to estimate metabolism. Finally, most in the field now prefer use LC-MS/MS for metabolite screening. The NMR analysis is still OK, but it is important to include estimates of the metabolite levels - mol/g wet weight – to compare with the literature.

3 An important interpretational aspect is in the model that the authors propose. They suggest that there is causality from the change in promoter methylation to SDH activity to succinate levels and on to effects on HIF1a. This is a nice and elegant model. (By the way this does make the assessment of SDH activity in the human tissue samples essential). More importantly, it could be that the changes in methylation are a secondary consequence of changes in succinate that then inhibits the 2-oxoglutarate-dependent dioxygenases (e.g. Jumonji) that are involved in DNA demethylation? The value of their data is still there and this possible should be discussed. It could be that in normal pregnancy that succinate is elevated by other mechanisms (enhanced glutaminolysis seems likely to me) and then leads to secondary changes in epigenetic marks.

4 In the summary the description of the changes in methylation/SDH/succinate in normal pregnancy seems to have got mixed up and is written as opposite to what the authors want to say – unless I've misunderstood something?

Reviewer #3:

Remarks to the Author:

This paper by Wang et al investigates the possible causes of Recurrent Spontaneous Abortion (RSA) in women. This is an important question, the findings are original and interesting. The paper is well written; the data are convincing and well presented. If it takes into account some experimental questions presented below, I feel this paper has good publication potential.

Summary of the data:

Using human samples and model cell lines, the authors:

1-carry out a metabolomic analysis of healthy vs. RSA villi. They report lower succinate and higher fumarate in RSA samples. A dozen other metabolites (AA or glucose metabolites) are similar between the two conditions, ruling out major artifacts.

2-SDHA and SDHB, two succinate dehydrogenases that consume succinate and produce fumarate are more expressed in RSA than in control villi. This is seen at RNA and protein level. A dozen other enzymes involved in glycolysis or the Krebs cycle are not differently expressed. In two different placental cell lines, increasing SDHB decreases succinate, while knocking down SDHB increases succinate.

3-decidua and villi have differing metabolomes and differing expression profiles of the glucose metabolic enzymes. These patterns also change over time. Early on (before day 60), SDHB is lower in villi than in decidua. Later on, the levels equalize. The succinate levels follow the inverse trend (high in villi early on, low later on).

4-some CpGs in the SDHB promoter are demethylated with time in normal samples. They are also demethylated early in RSA samples, possibly accounting for increased SDHB expression.

5-this region can be bound by MBD1 in vitro, and maybe also in cells. Knocking down MBD1 in cell lines increases the SDHB mRNA.

6-this same region contains a potential c-fos binding site, validated in EMSA, and maybe in ChIP as well. There is a possible competition between MBD1 and c-Fos binding to the region.

7-supplying extra succinate to placental cell lines stabilizes HIF1 and promotes invasion, reproducing the effect that is well-known in cancer. In normal villi, HIF1 is stabilized, but it is not stabilized in RSA villi.

8-a key question is whether the 2-fold succinate decrease seen in RSA may be sufficient to decrease HIF1 activation and alter placental function. This is addressed by injecting glycine into pregnant mice, which causes succinate utilization, and decreases the concentration 2-fold. This is sufficient to cause embryo resorption and is rescued by succinate complementation.

Main points

I like the fact that the paper addresses an important question using an excellent sample collection, in an unbiased manner, and arrives at original and interesting conclusions that are of obvious relevance to human health. I have some comments that could increase the quality of the paper further yet.

-can the authors precise how they measure or exclude contamination of the villi by the decidua and vice versa?

-I am surprised not to see any western blots on the JEG3 and HTR8. What is the degree of SDHB overexpression and knockdown in figure 1? Does 5-aza increase the protein level? What about MBD1 knockdown?

-the "semi-quantitative" ChIP assays of figure 3J-K are in fact not quantitative at all. It is a clear overinterpretation to claim less MBD1 binding and more c-Fos binding in the one RSA sample compared to the one control. ChIP-qPCR, even limited to cell lines, would be a vast improvement. Why is only one cell line used? Does the other not respond? Should fos binding go up upon MBD1 siRNA?

-there is some disconnect between figures 3A-D and figures S2 C-D. The methylation values for distal sites in figure 3 are relatively high, as expected (0.6 or more). In figure S2, the values rarely go over 0.3. What could cause this?

-the experiment in mice (Figure 5) is very interesting. But do the resorbed placenta have a phenotype similar to RSA?

-a schematic conclusion would help the readers. It would be nice to include a model summarizing succinate levels, SDHB expression, in vili vs decidua, early and late, in healthy vs RSA.

Reviewer #4:

Remarks to the Author:

Review of "Low embryonic villous succinate accumulation increases recurrent spontaneous abortion risk" by Wang and coworkers. This is an interesting study on a difficult and important problem of spontaneous abortion. The authors combine many technologies, including metabolomics, transcript and protein analysis, and functional testing in cells.

The problem is that it is lacking many fundamental statistical basics, and I question its reproducibility due to lack of statistical rigor. This is just a partial list of problems:

- 1) The actual sample number is unclear for each group for each assay. There is just a broad statement at the start about numbers of tissues, but these need to be specified clearly for each quantitative experiment.
- 2) The NMR metabolomics data is seriously lacking. How did the authors identify metabolites? How did they quantify metabolites? What is the confidence level of their analysis? There is no 2D NMR, and I am extremely skeptical of extracting all of the assignments from just 1D data. There is no description of how the authors statistically analyzed the data. They state that the experiments were "normalized" to TSP. This is totally incorrect. The data are referenced to TSP, but that does not normalize. Normalization is correcting the amount of signal to account for differences in sample quantity from sample to sample.
- 3) All of the statistical analysis is lacking. The authors use p-values from Student's t-tests that are not corrected for multiple sampling. When some form of multiple sampling correction is applied, the significance will be much lower.
- 4) All data should be deposited onto public databases so that others can verify the results directly.

POINT-BY-POINT RESPONSE TO THE REVIEWER

Reviewer #1 (Remarks to the Author):

In this manuscript the authors demonstrated that the decreased succinate in villi was responsible for increased RSA risk, and DNA methylation in SDHB promoter led to decreased SDHB expression which contributed to the decreased succinate levels. The authors utilized in vitro cells and mice models to reveal the essential roles of succinate in increasing trophoblast invasion/proliferation and preventing pregnancy loss. Overall, this manuscript is well organized. However, in this manuscript, the authors are somewhat exaggerating their findings and additional experiments are needed to confirm the authors' conclusion. Moreover, language editing would be helpful. Specific comments are provided below.

Response: We thank the Reviewer for this positive assessment of our study. Following the Reviewer's suggestions, we have provided additional evidence by experiments and the detailed response are listed in point-by-point responses below. We have also edited the language using the Elsevier Language Editing service.

1. In the Summary (Lines 30-31), the authors are overinterpreting their findings about the role of high succinate for embryo implantation. Although there is a correlation between first trimester of pregnancy and high succinate, without evidence showing the increased succinate accompanied with increased trophoblast invasion/proliferation during the first trimester of pregnancy, the conclusion regarding the essential role of high succinate for embryo implantation should be drawn with caution.

Response: We thank the Reviewer for this excellent suggestion. We have tuned down the conclusion and rephrased the sentence in the Summary part as follows: "During normal pregnancy, from the first to third trimesters, SDHB DNA methylation decreased, SDHB expression increased, and succinate levels decreased in villi from normal pregnant, indicating that sustained high level SDHB DNA methylation, low SDHB expression, and high succinate

levels in early stage pregnancy are correlated with successful embryo implantation” (Summary, Lines 26-30). Moreover, we discussed this limitation as “Although we could not obtain the direct evidence that increased succinate accompanied with increased trophoblast invasion/proliferation during the first trimester of pregnancy because of the limitation of sample collection, these findings emphasize the importance of succinate accumulation during the first trimester...” in Discussion section (page 15, lines 310-314).

2. Line 48, please revise “in the maternal fetal-interface” to “at the maternal-fetal interface”. Overall, the language of this manuscript needs to be further refined.

Response: We thank the Reviewer for this excellent suggestion. We have changed the description as Reviewer suggested (page 3, line 46). We have also edited the language of the manuscript using Elsevier Language Editing service.

3. Line 71, the role of hypoxia in regulating EVT differentiation has been described previously, thus the authors’ statement is inaccurate, especially previous publications have reported the inhibitory effect of hypoxia on EVT invasion.

Response: We thank the Reviewer for the instructive criticism. The role of hypoxia plays in regulating EVT differentiation does have been studied broadly, although inconsistent findings and conflicting views are still being debated. To provide a more balanced and complete view of this issue, we have reconstructed our introduction with extra references.

To our knowledge, this paradox of hypoxia can be partly explained by taking into consideration developmental stages and their chronological physiology. During the first trimester of pregnancy, the human placenta develops in a hypoxic environment caused by the occlusion of uterine spiral arterioles by extravillous trophoblasts (EVT). Before week 12 of pregnancy, relative low oxygen tension ($pO_2 < 20$ mmHg, oxygen concentration between 3 to 5% within intervillous space (IVS) O_2)¹ is crucial for successful pregnancy and can promote EVT invasion and differentiation.² Once the vascular remodeling is accomplished, oxygen

becomes vital for normal development. Hypoxia at this stage, which can most often happen with preeclampsia after week 20, does affect placenta function.

In our models, we found that during the first trimester of pregnancy, succinate acted as a pro-inflammation signal independent of oxygen, which means that during the first trimester of pregnancy, inflammation signals that increasing trophoblasts invasion ability could be dynamically regulated in the relative low tension environment.

On the basis of your suggestion, we have re-written the statement “however, it remains unclear whether and how hypoxia regulates extravillous trophoblasts” as “During the first trimester of pregnancy, the human placenta develops in a hypoxic environment caused by the occlusion of uterine spiral arterioles by extravillous trophoblasts. Before week 12 of pregnancy, relative low oxygen tension is crucial for successful pregnancy and can promote EVT invasion and differentiation. Once the vascular remodeling is accomplished, oxygen becomes vital for normal development. Hypoxia at this stage, which can most often happen with preeclampsia after week 20, does affect placenta function. Although it is well established that hypoxia plays a role in regulating extravillous trophoblasts function, however, the underlying mechanism remains unclear” in the Introduction part (page 4, Lines 68-75).

4. Line 96, the sentence should be “that metabolic alterations in villi is correlated with the occurrence of RSA,” rather than “contribute toward”.

Response: We thank the Reviewer for this excellent suggestion. We have changed the description as Reviewer suggested (page 5, lines 100-101).

5. Results: the authors need to state if the clinical samples used for screening from normal pregnancies and RSA patients are paired at gestational age.

Response: We thank the Reviewer for the suggestion. The normal pregnancies of first trimester and RSA patients were paired by gestational age. We have supplemented this statement “The gestational age of RSA patients and first-trimester normal pregnancies were paired” in the

revised Online only methods (Online only methods, page 1, line 12-13).

6. Line 141, “are required”, overexaggerating the findings again.

Response: We thank the Reviewer for the suggestion and tuned down the claim. We have used “are important for maintaining embryo implantation” instead of “are required to maintain embryo implantation” in the revised manuscript (Page 8, lines 148-150).

7. Line 207, “HIF-1a is known to promote cell invasion”, reference is required. “

Response: We thank the Reviewer for the suggestion and have added the reference 18 and 27 (page 11, line 226).

8. Lines 232-246, to directly demonstrate the essential role of succinate in preventing RSA occurrence, the supplementation of succinate to RSA mice models would be helpful. Moreover, instead of using glycine to decrease succinate expression *in vivo*, SDHB DNA methylation or SDHB overexpression *in vivo* would be preferred to be carried out followed by embryo-resorption rate and succinate level measurements, which could provide straightforward *in vivo* evidence.

Response: We thank the Reviewer for this great suggestion. To investigate the influence of succinate on outcome of pregnancy *in vivo*, following the reviewer’s suggestion, we have constructed a typical spontaneous abortion mouse model by mating CBA/J female mouse with DBA/2 male mouse, and used the CBA/J female mouse mating with BALB/C male mouse as a normal pregnancy mice control. The embryo-absorbing rate was 25% in CBA/J×DBA/2 and 6% in CBA/J×BALB/C models. When we supplemented succinate in CBA/J×DBA/2 model, the embryo-absorbing rate decreased significantly, from 25% decrease to 12% (Revised Fig. 5F).

CBA/J × DBA/2 mouse has been widely used as an abortion-prone model in an attempt to mimic RSA with characteristically high fetal resorption rate and normal karyotype. The

pregnancy-related abnormalities found in this model may be due to dysregulation of maternal immunity. A main manifestation of this immunity dysregulation is abnormal inflammatory cytokines levels, including increased TGF-1 β and decreased IL-1 β level.³ During early pregnancy, inflammatory cytokines such as IL-1 β play key roles in promoting extravillous trophoblasts invasion and migration.⁴ Succinate is known as an inflammatory metabolite that induces IL-1 β through HIF-1 α pathway.⁵ We found although the CBA/J \times DBA/2 mouse model was not accompanied by SDHB overexpression and succinate deficiency (revised Supplementary Fig. 4A, B), additional supplementation of dimethyl-succinate (DMS) increased IL-1 β level in mice villi tissues (Revised Fig. 5E). Thus, these results indicated that increased succinate decreased spontaneous abortion risk by accumulating IL-1 β levels in CBA/J \times DBA2 model.

In the revised manuscript, the description of this result “Moreover, in the CBA/J \times DBA2 spontaneous abortion mouse model, increasing pregnant mice succinate levels via intraperitoneal succinate injection increased IL-1 β levels in mouse villi samples (Fig. 5E), and significantly decreased the embryo-resorption rate (Fig. 5F)” has been provided in last section in Result section (page 13, lines 262-265). This result was also discussed in Discussion section as “We found although the CBA/J \times DBA/2 spontaneous abortion mouse model was not accompanied by SDHB overexpression and succinate deficiency in villi tissue (Supplementary Fig. 4A, B), additional supplementation of DMS further increased IL-1 β level in mice villi tissue and decreased the embryo-resorption in this spontaneous abortion mouse model” (page 15, lines 306-310). The method information has been provided in revised Online only methods (Online only methods, page 12, lines 246-250).

Moreover, we acknowledged that in addition to administration of glycine, a SDHB overexpression mouse model was helpful to elucidate our model. We have tried to use DNA methylation inhibitors, decitabine and azacitidine, to decrease SDHB methylation and increase SDHB expression. However, although DNA methylation inhibitors were able to decrease succinate levels in cultured cells, they showed strong toxicity to embryos and led to resorption of all the embryos (we checked the embryo resorption in E14.5). Besides, we also tried to construct the placenta-specific SDHB overexpression mouse embryo model by using a

trophoblast-specific lentiviral gene transfer method;⁶ unfortunately, we could not obtain this mouse model. We have discussed this limitation in Discussion section as “Besides, a SDHB overexpression mouse model is needed to provide the *in vivo* evidence that low succinate induces embryo-resorption in future studies” in the Discussion part (Page 14, lines 286-287).

9. In Discussion, the shortages of this study should be mentioned and discussed.

Response: We thank the Reviewer for this great suggestion. In the revised Discussion section, we discussed several limitations of this study, including the lack of evidence showing that the increased succinate accompanies the increased trophoblast invasion/proliferation during the first trimester of pregnancy (page 15, lines 310-313), the uncovered causality between succinate levels and SDHB methylation (page 14, line 281-286), and the lack of SDHB overexpression mouse model (page 14, lines 286-287).

10. Methods: How long the JEG3 and HTR8 were treated with succinate followed by cell invasion and proliferation assays?

Response: We apologize for not having provided the detailed method information. In the cell invasion assay, we treated cultured JEG3 and HTR8 cells with 0.5 mM or 2 mM DMS for 2 hours before we performed the invasion assays. After seeding the cells in the upper part of the chamber, 0.5 mM and 2 mM DMS were added to the medium of the lower chamber, respectively. We modified this description in revised Online only methods part (Online only methods, pages 7-8, lines 153-156). In the proliferation assays, we cultured cells with the DMEM + 10% FBS medium, containing 2 mM DMS or not. The cultured medium was changed daily until the cells were harvested. We have supplemented the information in the revised Online only Methods section (Online only methods, page 8, lines 165-168).

11. Figure 5C: Please provide the representative photo for embryo-resorbing rate.

Response: We thank the Reviewer for this suggestion and have provided the representative image for embryo-resorbing rate in revised Fig. 5C.

Reviewer #2 (Remarks to the Author):

Overview

I found this to be a fascinating paper that contained an interesting idea that both impacts on our understanding of the role of metabolite signalling in embryo genesis. As the authors are working with human tissue the experiments that are possible are obviously circumscribed. Furthermore, as with all ideas that are breaking new ground there are many more experiments that could be done. I have a number of technical quibbles, but again these may not be possible to address given the tissue samples used. While I am generally supportive of this paper there are a few points I'd like to see addressed with more experimentation before publication. In addition, there are some interpretational issues that I feel could be clarified.

Response: We thank the Reviewer for this positive assessment of our study. Following the reviewer's suggestion, we have provided additional evidence by experiments and the details are listed in point-by-point responses below.

Major points

1 The role of SDHB and its changes in expression and protein levels in RSA are well supported by the data. However, the relationship between expression of one of four subunits of SDH and its activity is not clear. For the hypothesis being assessed the activity of SDH is key and has to be measured. This should be done on human tissue homogenates and normalised to citrate synthase activity. It would also be useful to compare mtDNA levels to marker nuclear genes. In addition, it is straightforward to assess SDH activity on histological samples. This can also be done relative to cytochrome oxidase. These assays would add a lot and in my view are

essential.

Response: We thank the Reviewer for these great suggestions. Following the suggestions of reviewer, we have assessed the activities of SDH in villous and decidua samples from either 10 RSA patients or 10 healthy controls. The SDH activity was measured by SDH activity kit (K660-100, Biovision) and normalized to the citrate synthase activity measured by CS activity kit (K287-100, Biovision). We found the SDH relative activities increased remarkably in RSA villous tissues, but not in decidua tissues, compared to the healthy controls. We put this result in revised Fig. 1G, and added the description “Moreover, we confirmed that the relative SDH activity increased remarkably in RSA villous tissues, but not in decidua tissues, compared to those in the healthy controls (Fig. 1G)” in the first section of Result part (page 6, lines 116-118). The relative method was provided in the revised Online only methods (Online only methods, page 10, lines 205-207).

Moreover, by evaluating the mitochondrial mass through measuring the ratio of mitochondrial DNA to nucleic DNA, we found either overexpression or knockdown of SDHB did not alter the mitochondrial mass in both JEG3 and HTR8 cells. This result indicated that enhanced SDHB expression decreases succinate levels through increasing intracellular SDH activity. We provided this result in Supplementary Fig. 1B and described this data as “Together with the finding that either SDHB overexpression or SDHB knockdown did not change the mitochondrial mass in JEG3 and HTR8 cells (Supplementary Fig. 1B), these results indicate that enhanced SDHB expression decreases succinate levels which correlate with RSA onset through increasing intracellular SDH activity” in the Results section (page 6, lines 118-121). The method of mitochondrial mass measurement was provided in the revised Online only methods (Online only methods, page 4, lines 68-71).

2 I have some technical concerns about the way in which the metabolomics was done. For metabolites, most especially succinate, there can be extensive changes in tissues post harvesting. For many of the studies in my lab we clamp freeze the tissue at liquid nitrogen temperature using Wollenberger clamps. Depending on the physiological context, a few

seconds where the tissue is not perfused but stays warm can be critical. Now, I recognize that given the constraints in which these samples were analysed this may be challenging. Even so, it is essential that a full description and timings of how the samples were obtained, cooled and stored is given. Limitations to the approach used should be discussed – and the important thing is that succinate seems to change selectively between control and RSA.

Response: We thank the Reviewer for the suggestion. We acknowledged that high quality samples are important to the metabolite's quantification. So, we shortened the time costs from samples harvesting as much as possible. In detail, after curettage, the tissues were immediately collected under sterile conditions into pre-chilled PBS and divided into villi and decidua. Then the villus and decidual tissues were washed again with pre-chilled PBS to exclude contamination of the villi by the decidua and vice versa. The samples were dried with paper towels and cut into small pieces. For metabolites analysis, samples were fixed by adding 5 volumes of pre-chilled 70% methanol and stored at -80 °C until use. For western blotting, mRNA extraction, or DNA extraction, samples were frozen in liquid nitrogen, and stored in the refrigerator at -80 °C until use. The entire sample collection procedure took less than 2 minutes.

In the revised manuscript, we newly collected 30 paired villi and decidua samples from RSA and controls using the same sampling procedure for absolute quantification of metabolites. The absolute quantification results of metabolites were in accordance with our previous relative quantification results of metabolites, suggesting that the sampling approach was stable.

We have provided this detailed information in the revised Online only methods section (Online only methods, pages 1-2, lines 16-23).

A related issue is the way in which the samples are just homogenised in phosphate buffer. Usually in this field frozen samples are first rapidly homogenised in a denaturing solvent – e.g. acid, acetonitrile, methanol etc – to ensure that there is no enzymatic activity during resuspension. Again, it may be challenging to do this. But even so, a control in which mock extractions of ^{13}C -succinate \pm tissues are carried out to estimate metabolism. Finally, most in the field now prefer use LC-MS/MS for metabolite screening. The NMR analysis is still OK, but

it is important to include estimates of the metabolite levels - mol/g wet weight – to compare with the literature.

Response: We thank the Reviewer for the question, comment, and suggestion.

We apologize for not having provided the detailed sample preparation method information in the NMR analysis. In the metabolite's measurement, villi or decidua tissues (about 50 mg) were extracted with 600 μ L of precooled methanol-water mixture (2/1, v/v) using a tissue lyser (QIAGEN TissueLyser II, Germany). Supernatant for each sample was collected respectively after 10 min of centrifugation ($11180 \times g$, 4 °C). Such extracting procedure was further repeated twice. Three supernatants obtained for each sample were combined and centrifuged ($16099 \times g$, 4 °C) for another 10 min. The resultant supernatant for each sample was respectively lyophilized after removal of methanol in vacuo. The dried extracts were reconstituted into 600 μ L of phosphate buffer (0.15 M, K_2HPO_4 - NaH_2PO_4 , pH 7.43) containing 80% D_2O (v/v) and trimethylsilyl propionate (0.2915 mM). The mixture was then centrifuged at $16099 \times g$ for 10 min at 4 °C. Then, 550 μ L each supernatant was transferred into a standard 5 mm NMR tube for analysis. We provided this information in the revised Online only Methods section (Online only Methods, page 4, lines 74-83).

Moreover, following the great suggestion of reviewer, we have measured the absolute concentrations of metabolites in human villi and decidua samples. To obtain the absolute quantification of metabolites, we newly collected 30 pairs of villi and decidua samples and measured the metabolites concentrations using NMR. The new absolute metabolites concentration results (revised Tables 1, 2, and Supplementary Table 1) were in accordance with previous relative metabolites concentration results presented in the original manuscript (original Fig. 1A, 2C; Supplementary Fig. 1B, C). Moreover, we found that the metabolites levels were comparable to the results in literature. For example, the concentration of succinate was 0.43–0.59 μ mol/g in decidua and villi tissues, which was similar to the reported concentration in cerebral (nearly 0.5 μ mol/g).⁷⁻⁹ The concentration of taurine was 3.8–4.4 μ mol/g in decidua and villi tissues, which was similar to that of a previous study (3.5 μ mol/g) in term human placentas.¹⁰ The concentration of lactate was 2.9–3.4 μ mol/g in decidua and villi tissues, which

was similar to the concentration in a previous study in trophoblast cells (nearly 3–6 $\mu\text{mol/g}$).¹¹

In the revised manuscript, we have provided the absolute metabolites levels in tissue in revised Table 1, 2, and Supplementary Table 1, and the previous relative quantification results have been removed. We have described the new result in Result section (page 5, lines 96-101; page 7, lines 137-139). The source data are provided with this paper in Source data file 1. The NMR raw data are deposited in the MetaboLights database (www.ebi.ac.uk/metabolights/MTBLS2143) with identifier “MTBLS2143”. Because the submitted NMR raw data will not be made available to the world until MetaboLights completed the online validation (minimum 28 days is required for our validation and curation), we also uploaded the NMR raw data in the dropbox online (<https://www.dropbox.com/sh/u9kw2ksw3p77t7y/AACjyq0NAn6uY5yhHbcWnih9a?dl=0>). **So, please check the NMR raw data in Dropbox before MetaboLights completing the validation.** The detailed NMR and metabolites quantification method are provided in the revised Online only Methods (Online only Methods, pages 4-6, lines 73-122).

3 An important interpretational aspect is in the model that the authors propose. They suggest that there is causality from the change in promoter methylation to SDH activity to succinate levels and on to effects on HIF1 α . This is a nice and elegant model. (By the way this does make the assessment of SDH activity in the human tissue samples essential). More importantly, it could be that the changes in methylation are a secondary consequence of changes in succinate that then inhibits the 2-oxoglutarate-dependent dioxygenases (e.g. Jumonji) that are involved in DNA demethylation? The value of their data is still there and this possible should be discussed. It could be that in normal pregnancy that succinate is elevated by other mechanisms (enhanced glutaminolysis seems likely to me) and then leads to secondary changes in epigenetic marks.

Response: We thank the Reviewer for this great suggestion. Following the Reviewer's suggestion, we made the assessment of SDH activity in human tissue samples as we described earlier herein (in question 1). To investigate that the changes in methylation are a secondary

consequence of changes in succinate, we overexpressed either TET1 or TET2 individually in JEG3 and HTR8 cells, treated the cells with or without 2 mM DMS. After 24 hours treatment, we isolated genomic DNA from JEG3 and HTR8 cells and determined 5hmC levels by dot-blot, which allowed for quantitative measurement as we have previously reported.¹² These experiments demonstrated that although ectopic expression of either TET1 or TET2, resulted in high levels of 5hmC in both JEG3 and HTR8 cells, additional administration of DMS caused a substantial decrease of TET1/TET2-mediated 5hmC production (revised Fig 4B). These results indicated, beside elevation of HIF-1 α pathway, the 2-oxoglutarate-dependent dioxygenases activities were inhibited, and confirmed our notion that succinate competes with α -KG and leads to TET inhibition.

Moreover, we found that the DNA methylation levels of SDHB promoter increased (revised Fig. 4C) and SDHB expression levels decreased (revised Fig. 4D) in DMS treated cells. These results suggested the presence of a positive feedback regulation between SDHB methylation and succinate. We agreed with the reviewer's great opinion that in normal pregnancy succinate may be elevated by other mechanism. The underlying mechanism deserves further studies in the future. We have provided the description of this data in the revised Result section (page 11, lines 216-225), and described this point in revised Discussion section (page 14, lines 279-286). The dot-blot assay method was provided in the revised Online only Methods (Online only Methods, page 7, lines 137-144).

4 In the summary the description of the changes in methylation/SDH/succinate in normal pregnancy seems to have got mixed up and is written as opposite to what the authors want to say – unless I've misunderstood something?

Response: We thank the Reviewer for this question. We apologized for the misleading description in the Summary. In the first trimester of normal pregnancy, the villous tissue had high level SDHB DNA methylation, low level SDHB expression, and high level of succinate. From the first to third trimesters in the villi during normal pregnancy, SDHB DNA methylation decreased, SDHB expression increased, and succinate levels decreased. Together with the

facts that in RSA patients, villous samples displayed reduced SDHB DNA methylation, elevated SDHB expression, and reduced succinate levels, these results indicated that sustained high level of succinate in the first trimester is important for embryo implantation. We have modified the description as “During normal pregnancy, from the first to third trimesters, SDHB DNA methylation decreased, SDHB expression increased, and succinate levels decreased in villi from normal pregnant, indicating that sustained high level SDHB DNA methylation, low SDHB expression, and high succinate levels in early stage pregnancy are correlated with successful embryo implantation” (Summary part, lines 26-30).

Reviewer #3 (Remarks to the Author):

This paper by Wang et al investigates the possible causes of Recurrent Spontaneous Abortion (RSA) in women. This is an important question, the findings are original and interesting. The paper is well written; the data are convincing and well presented. If it takes into account some experimental questions presented below, I feel this paper has good publication potential.

Summary of the data:

Using human samples and model cell lines, the authors:

1-carry out a metabolomic analysis of healthy vs. RSA villi. They report lower succinate and higher fumarate in RSA samples. A dozen other metabolites (AA or glucose metabolites) are similar between the two conditions, ruling out major artifacts.

2-SDHA and SDHB, two succinate dehydrogenases that consume succinate and produce fumarate are more expressed in RSA than in control villi. This is seen at RNA and protein level. A dozen other enzymes involved in glycolysis or the Krebs cycle are not differently expressed. In two different placental cell lines, increasing SDHB decreases succinate, while knocking down SDHB increases succinate.

3-decidua and villi have differing metabolomes and differing expression profiles of the glucose

metabolic enzymes. These patterns also change over time. Early on (before day 60), SDHB is lower in villi than in decidua. Later on, the levels equalize. The succinate levels follow the inverse trend (high in villi early on, low later on).

4-some CpGs in the SDHB promoter are demethylated with time in normal samples. They are also demethylated early in RSA samples, possibly accounting for increased SDHB expression.

5-this region can be bound by MBD1 in vitro, and maybe also in cells. Knocking down MBD1 in cell lines increases the SDHB mRNA.

6-this same region contains a potential c-fos binding site, validated in EMSA, and maybe in ChIP as well. There is a possible competition between MBD1 and c-Fos binding to the region.

7-supplying extra succinate to placental cell lines stabilizes HIF1 and promotes invasion, reproducing the effect that is well-known in cancer. In normal villi, HIF1 is stabilized, but it is not stabilized in RSA villi.

8-a key question is whether the 2-fold succinate decrease seen in RSA may be sufficient to decrease HIF1 activation and alter placental function. This is addressed by injecting glycine into pregnant mice, which causes succinate utilization, and decreases the concentration 2-fold. This is sufficient to cause embryo resorption and is rescued by succinate complementation.

Response: We thank the Reviewer for these positive comments.

Main points

I like the fact that the paper addresses an important question using an excellent sample collection, in an unbiased manner, and arrives at original and interesting conclusions that are of obvious relevance to human health. I have some comments that could increase the quality of the paper further yet.

Response: We thank the Reviewer for this comment.

-can the authors precise how they measure or exclude contamination of the villi by the decidua and vice versa?

Response: We thank the Reviewer for this question. We generally distinguish villi and decidual tissue by appearance because the villus and decidual tissues are totally different in their appearance. First, the villus tissues are like white snowflakes and they can float up in the water. However, the decidual tissues are light pink and flaky. They can sink in the water. So, during PBS washes of the tissues, we can differentiate them by their appearances and correctly select the appropriate tissue. Second, there is absolutely no decidual cells among villous tissues. Third, contamination of little villous cells may be present only at the implantation site, which has little effect on the result. Thus, we can exclude contamination of the villi by the decidua and vice versa during the collection of the samples.

-I am surprised not to see any western blots on the JEG3 and HTR8. What is the degree of SDHB overexpression and knockdown in figure 1? Does 5-aza increase the protein level? What about MBD1 knockdown?

Response: We thank the Reviewer for these great suggestions. We apologized for lack of western blots examination of protein expression on the JEG3 and HTR8. We have provided the western blot validation of SDHB expression in the revised Fig. 1C and Fig. 1E. Besides, we also provided the knockdown efficiencies for MBD1/2/3/4/5/6 and MECP2 in revised Supplementary Fig. 3C.

We examined the protein levels of SDHB and CS in JEG3 and HTR8 cells, treated with either decitabine/azacitidine or not. The results showed that the protein levels of SDHB, but not CS, increased in decitabine/azacitidine treated cells (revised Fig. 3G). We described this result in the revised Result section (page 9, lines 172-173).

Moreover, we confirmed that SDHB protein level increased in MBD1 knockdown cells (revised Fig. 3I). We described this new data in the Result section (page 9, lines 185-188).

-the “semi-quantitative” ChIP assays of figure 3J-K are in fact not quantitative at all. It is a clear overinterpretation to claim less MBD1 binding and more c-Fos binding in the one RSA sample compared to the one control. ChIP-qPCR, even limited to cell lines, would be a vast improvement. Why is only one cell line used? Does the other not respond? Should fos binding go up upon MBD1 siRNA?

Response: We thank the Reviewer for these questions. Following the Reviewer’s suggestions, we have improved this section as follows. First, using the ChIP-qPCR, we found the binding ability of c-Fos to SDHB promoter increased significantly in the MBD1 knockdown JEG3 and HTR8 cells (revised Fig. 3M). Second, for the result in the original Fig. 3K (revised Fig. 3N), we had measured the binding affinities of MBD1 and c-Fos to SDHB promoter region in villous samples from three RSA and three healthy controls. The original Fig. 3K (revised Fig. 3N) showed the representative results. As an improvement, we have used ChIP-qPCR to quantify the binding affinities of MBD1 and c-Fos to the SDHB promoter. The results showed decreased MBD1 and increased c-Fos bound to the SDHB promoter in villi from RSA patients, compared to the villi from healthy controls (revised Fig. 3N). We have updated the description of ChIP assay results in the revised Result section (page 10, lines 196-202).

-there is some disconnect between figures 3A-D and figures S2 C-D. The methylation values for distal sites in figure 3 are relatively high, as expected (0.6 or more). In figure S2, the values rarely go over 0.3. What could cause this?

Response: We thank the Reviewer’s for this question. Interestingly, we found the DNA methylation levels varied in the promoter region. In detail, the CpG sites in distal promoter were highly methylated. In contrast, the methylation levels of CpG sites in the middle promoter region were lower than in the distal promoter. The methylation levels of CpG sites in the proximal promoter were very low. Because the difference of methylation levels in the SDHB promoter were huge, we have showed the results in three separate figures to facilitate the reading of methylation changes between RSA and control groups. All the relative source data are provided as Source Data file 2.

-the experiment in mice (Figure 5) is very interesting. But do the resorbed placenta have a phenotype similar to RSA?

Response: We thank the Reviewer for this question. There are similarities and differences. (1) Similarities: During the early pregnancy, RSA patients undergo abortion as the result of growth retardation or even the stoppage of growth of the fetus. At the same time or later on, women developed abdominal pain and vaginal bleeding. Then, the embryonic tissues can be partially or totally removed from women's body. Consistently, the pregnant mice develop embryo absorption accompanied with the embryonic tissues stopping growing and becoming black and hard. (2) Differences: in our mice model, the resorbed placenta was not accompanied with vaginal bleeding and early embryo delivery.

-a schematic conclusion would help the readers. It would be nice to include a model summarizing succinate levels, SDHB expression, in villi vs decidua, early and late, in healthy vs RSA.

Response: We thank the Reviewer for this great suggestion. We have provided the schematic model summarizing succinate levels, SDHB expression, in villi vs decidua, early and late, in healthy vs RSA, in the revised Fig. 5G.

Reviewer #4 (Remarks to the Author):

Review of "Low embryonic villous succinate accumulation increases recurrent spontaneous abortion risk" by Wang and coworkers. This is an interesting study on a difficult and important problem of spontaneous abortion. The authors combine many technologies, including metabolomics, transcript and protein analysis, and functional testing in cells.

The problem is that it is lacking many fundamental statistical basics, and I question its

reproducibility due to lack of statistical rigor. This is just a partial list of problems:

1) The actual sample number is unclear for each group for each assay. There is just a broad statement at the start about numbers of tissues, but these need to be specified clearly for each quantitative experiment.

Response: We thank the Reviewer for this great suggestion. We have supplemented detailed sample information in revised manuscript and figure legends. Moreover, we have provided all the source data in Source data files, including the relevant data for all Tables and Supplementary Tables (Source data file 1), relevant data for all Figures and Supplementary Figures (Source data file 2), and all the uncropped gels and blots for all Figures and Supplementary Figures (Source data file 3).

2) The NMR metabolomics data is seriously lacking. How did the authors identify metabolites? How did they quantify metabolites? What is the confidence level of their analysis? There is no 2D NMR, and I am extremely skeptical of extracting all of the assignments from just 1D data. There is no description of how the authors statistically analyzed the data. They state that the experiments were “normalized” to TSP. This is totally incorrect. The data are referenced to TSP, but that does not normalize. Normalization is correcting the amount of signal to account for differences in sample quantity from sample to sample.

Response: We apologize for having provided the misleading and less detailed method information. The statement that the experiments were “normalized” to TSP was incorrect. In fact, the data are first divided by TSP and then normalized to samples' wet weight, the TSP was used only for calibration. In order to identify metabolites, a series of 2D NMR spectra were acquired and processed, including ^1H - ^1H correlation spectroscopy (COSY), ^1H - ^1H total correlation spectroscopy (TOCSY), J-resolved spectroscopy (JRES), ^1H - ^{13}C heteronuclear single quantum correlation (HSQC), and ^1H - ^{13}C heteronuclear multiple bond correlation (HMBC) 2D NMR spectra. The metabolites were assigned on the basis of literature data, and further individually confirmed with 2D NMR data including COSY, TOCSY, JRES, HSQC and HMBC (Pictured below). We have provided the two-dimensional (2D) NMR spectra of

metabolites in the Supplementary Fig. 1A. The (2D) NMR spectra raw data are deposited in the MetaboLights database (www.ebi.ac.uk/metabolights/MTBLS2143) with identifier “MTBLS2143”. Because the submitted NMR raw data will not be made available to the world until MetaboLights completed the online validation (minimum 28 days is required for our validation and curation), we also uploaded the NMR raw data in the dropbox online (<https://www.dropbox.com/sh/u9kw2ksw3p77t7y/AACjyq0NAn6uY5yhHbcWnih9a?dl=0>). **So, please check the NMR raw data in Dropbox before MetaboLights completing the validation.**

In order to quantify metabolites, one ^1H NMR spectrum was acquired for each tissue extract with a standard NOESYGPPR1D pulse sequence ($\text{RD}-G_1-90^\circ-t_1-90^\circ-t_m-G_2-90^\circ-\text{acq}$) with the recycle delay (RD) of 2 s and t_m of 100 ms. The total relaxation delay time was 26 s which allowed the completely relaxed NMR spectra to be obtained. All the NMR spectra were processed using the software package TOPSPIN (V3.6.0, Bruker Biospin, Germany). For ^1H NMR spectra, an exponential window function was employed with a line broadening factor of 1 Hz and zero-filled to 128 k prior to Fourier transformation. Each spectrum was then phase- and baseline-corrected manually with the chemical shift referenced to TSP (δ 0.00). The spectral regions were then integrated into bins with the width of 0.002 ppm (1.2 Hz) using AMIX software package (V3.8.3, Bruker Biospin). The absolute concentration of metabolites was calculated with the known concentration of TSP.

In the revised vision, we measured the absolute concentrations of metabolites from newly collected 30 pairs of villi and decidua samples, and found that the metabolites levels were comparable those reported in literature. For example, the concentration of succinate was 0.43–0.59 $\mu\text{mol/g}$ in decidua and villi tissues, which was similar to the reported concentration in cerebral (nearly 0.5 $\mu\text{mol/g}$).⁷⁻⁹ The concentration of taurine was 3.8–4.4 $\mu\text{mol/g}$ in decidua and villi tissues, which was similar to that of a previous study (3.5 $\mu\text{mol/g}$) in term human placentas.¹⁰ The concentration of lactate was 2.9–3.4 $\mu\text{mol/g}$ in decidua and villi tissues, which was similar to the concentration in a previous study in trophoblast cells (nearly 3–6 $\mu\text{mol/g}$).¹¹ We have provided the methods we used to quantify metabolites in detail in the revised Online only methods (Online only Methods, pages 4-6, lines 73-122). In addition, we have deposited

all the NMR raw data in the MetaboLights database (www.ebi.ac.uk/metabolights/MTBLS2143) with identifier “MTBLS2143”.

3) All of the statistical analysis is lacking. The authors use *p*-values from Student’s *t*-tests that are not corrected for multiple sampling. When some form of multiple sampling correction is applied, the significance will be much lower.

Response: We thank the Reviewer for this great suggestion. In the revised version, we used Benjamini & Hochberg method to correct the *p* values of the *t*-test for metabolites involving multiple sampling, and the *p*-values obtained after correction did not affect the conclusion in this study. We have updated this information in revised Online only methods (Online only Methods, page 12, lines 259-260).

4) All data should be deposited onto public databases so that others can verify the results directly.

Response: We thank the Reviewer for this great suggestion. We have provided all the source data in Source data files, including the relevant data for all Tables and Supplementary Tables (Source data file 1), relevant data for all Figures and Supplementary Figures (Source data file 2), and all the uncropped gels and blots for all Figures and Supplementary Figures (Source data file 3). Moreover, we have deposited all the NMR raw data in the MetaboLights database (www.ebi.ac.uk/metabolights/MTBLS2143) with identifier “MTBLS2143”. (Please check the NMR raw data in Dropbox before MetaboLights completing the validation: (<https://www.dropbox.com/sh/u9kw2ksw3p77t7y/AACjyq0NAn6uY5yhHbcWnih9a?dl=0>))

Reference in *Point-by-point response to the Reviewer*

1. Ottosen LDM, Hindkaer J, Husth M, et al. Observations on Intrauterine Oxygen Tension Measured by Fibre-Optic Microsensors. *Reprod Biomed Online*. 2006;13(3):380-5.
2. Treissman J, Yuan V, Baltayeva J, et al. Low Oxygen Enhances Trophoblast Column Growth by Potentiating Differentiation of the Extravillous Lineage and Promoting LOX Activity. *Development*. 2020;147(2): dev181263.
3. Yi X, Zhang J, Liu H, et al. Suppressed Immune-Related Profile Rescues Abortion-Prone Fetuses: A Novel Insight Into the CBA/J × DBA/2J Mouse Model. *Reprod Sci*. 2019;26(11):1485-1492.
4. Prutsch N, Fock V, Haslinger P, et al. The role of interleukin-1 β in human trophoblast motility. *Placenta*. 2012;33(9):696-703.
5. Tannahill GM, Curtis AM, Adamik J, et al. Succinate is an inflammatory signal that induces IL-1 β through HIF-1 α . *Nature*. 2013;496(7444):238-242.
6. Okada Y, Ueshin Y, Isotani A, et al. Complementation of placental defects and embryonic lethality by trophoblast-specific lentiviral gene transfer. *Nat Biotechnol*. 2007;25(2):233-7.
7. Benzi G, Arrigoni E, Marzatico F, et al. Influence of some biological pyrimidines on the succinate cycle during and after cerebral ischemia. *Biochem. Pharmacol*. 1979;28:2545–2550.
8. Benzi G, Pastoris O, Dossena M. Relationships between gamma-aminobutyrate and succinate cycles during and after cerebral ischemia. *J. Neurosci. Res*. 1982;7:193–201.
9. Folbergrova J, Ljunggren B, Norberg K, et al. Influence of complete ischemia on glycolytic metabolites, citric acid cycle intermediates, and associated amino acids in the rat cerebral cortex. *Brain Res*. 1974;80:265–279.
10. Philipps AF, Holzman IR, Teng C, Battaglia FC. Tissue concentrations of free amino acids in term human placentas. *Am J Obstet Gynecol*. 1978;131(8):881-887.
11. Kay HH, Zhu S, Tsoi S. Hypoxia and lactate production in trophoblast cells. *Placenta*. 2007;28(8-9):854-860.
12. Xu W, Yang H, Liu Y, et al. Oncometabolite 2-hydroxyglutarate is a competitive inhibitor of α -ketoglutarate-dependent dioxygenases. *Cancer Cell*. 2011;19(1):17-30.

Reviewers' Comments:

Reviewer #1:

Remarks to the Author:

The authors have carefully revised the manuscript and resolved the majority of my concerns in the current version. However, I still have a question as follows:

Regarding the Online Only Methods (Line 14), please elaborate on the inclusion and exclusion criteria of RSA patients/villi, such as how the chromosomal abnormality of patients was examined and what kinds of immune and metabolic diseases were excluded, etc.. Moreover, have the authors checked the chromosomal abnormality of the villi? If yes, by which method? Since the majority of RSA is correlated with the chromosomal abnormality/genetic mutation of the villi, is it possible that the genetic mutations at the SDHB region are responsible for reduced SDHB DNA methylation/SDHB expression observed in RSA villi (is there any report regarding the relationship between SDHB region genetic mutation and early miscarriage)?

Reviewer #2:

Remarks to the Author:

None

Reviewer #3:

Remarks to the Author:

I am satisfied with the authors' answers to my questions. They have been addressed experimentally, for the most part, and the results are convincing.

Reviewer #4:

Remarks to the Author:

This revision is significantly improved. Congratulations on an interesting study!

Small edit: Line 28 should either be "pregnant women" or "pregnancy"

POINT-BY-POINT RESPONSE TO THE REVIEWER

Reviewer #1 (Remarks to the Author):

The authors have carefully revised the manuscript and resolved the majority of my concerns in the current version. However, I still have a question as follows:

Regarding the Online Only Methods (Line 14), please elaborate on the inclusion and exclusion criteria of RSA patients/villi, such as how the chromosomal abnormality of patients was examined and what kinds of immune and metabolic diseases were excluded, etc.. Moreover, have the authors checked the chromosomal abnormality of the villi? If yes, by which method? Since the majority of RSA is correlated with the chromosomal abnormality/genetic mutation of the villi, is it possible that the genetic mutations at the SDHB region are responsible for reduced SDHB DNA methylation/SDHB expression observed in RSA villi (is there any report regarding the relationship between SDHB region genetic mutation and early miscarriage)?

Response: We thank the reviewer for this excellent suggestion. We have elaborated the exclusion criteria of RSA patients in the revised version of Online only methods (page 1, lines 13-22). In fact, Patients who had experienced infection, endocrine or metabolic diseases, chromosomal abnormalities, anatomic abnormalities or immune diseases were excluded in this research. Besides, we sequenced *SDHB* promoter region in villi from 10 normal pregnant women and 20 RSA patients and found no genetic mutation in *SDHB* promoter region reported to be associated with abortion. We have added the description of RSA patients *SDHB* promoter region sequencing results in the revised version of our manuscript (page 8, lines 158-160). We also provided the detailed sequence result in Supplementary Table 2 and added related method in the revised version of Online only methods (page 2, lines 34-40). All raw sequencing data created in this study have been uploaded to the National Omics Data Encyclopedia (NODE; <https://www.biosino.org/node/project/detail/OEP001375>) with the accession number OEP001375.

Reviewer #2 (Remarks to the Author):

None

Response: We thank the reviewer for this positive assessment of our study.

Reviewer #3 (Remarks to the Author):

I am satisfied with the authors' answers to my questions. They have been addressed experimentally, for the most part, and the results are convincing.

Response: We thank the reviewer for this positive assessment of our study.

Reviewer #4 (Remarks to the Author):

This revision is significantly improved. Congratulations on an interesting study!

Small edit: Line 28 should either be "pregnant women" or "pregnancy"

Response: We thank the reviewer of for this positive assessment of our study. We have changed the description to "pregnancy" (Line 26) in the new version of manuscript.

Reviewers' Comments:

Reviewer #1:

Remarks to the Author:

All my concern have been addressed.